

# Towards European-Scale Convection-Resolving Climate Simulations

David Leutwyler[1], Oliver Fuhrer[3], Xavier Lapillonne[2,3], Daniel Lüthi[1], and
Christoph Schär[1]

[1]Institute for Atmospheric and Climate Science, ETH Zürich, Switzerland
[2]Center for Climate Systems Modeling C2SM, ETH Zurich, Switzerland
[3]Federal Office of Meteorology and Climatology, MeteoSwiss, Zürich, Switzerland

*Correspondence to:* David Leutwyler (david.leutwyler@env.ethz.ch)

**Abstract.** The representation of moist convection in climate models represents a major challenge, due to the small scales involved. Using horizontal grid spacings of O(1km), convection-resolving weather and climate models allow to explicitly resolve deep convection. However, due to their extremely demanding computational requirements, they have so far been limited to short simulations

and/or small computational domains. Innovations in supercomputing have led to new hybrid node designs, mixing conventional multicore CPUs and accelerators such as graphics processing units (GPUs). One of the first atmospheric models that has been fully ported to these architectures is the COSMO model.

Here we demonstrate the convection-resolving COSMO model on continental scales using a version of the model capable of using GPU accelerators. The verification of a week-long simulation

containing winter storm Kyrill shows that, for this case, convection-parameterizing simulations and convection-resolving simulations agree well. Furthermore we demonstrate the applicability of the approach to longer simulations by conducting a three-month long simulation of the summer season 2006. Its results corroborate the findings found on smaller domains such as more credible representation of the diurnal cycle of precipitation in convection-resolving models and a tendency to produce

more intensive hourly precipitation events. Both simulations also show how the approach allows for the representation of interactions between synoptic-scale and meso-scale atmospheric circulations at scales ranging from 1000 to 10 km. This includes the formation of sharp cold frontal structures, convection embedded in fronts and small eddies, or the formation and organization of propagating cold pools. Finally we assess the performance gain from using heterogeneous hardware equipped with

GPUs with respect to multi-core hardware. With the COSMO model, we now use a climate model





that has all the necessary modules required for real-case convection-resolving climate simulations on GPUs.

## 1 Introduction

The inadequate representation of clouds and moist convection represents a major challenge of state-of-the-art climate models (Stevens and Bony, 2013). An important component of the problem are the scale interactions between small-scale turbulent and convective processes at scales around and below 1 km, and large-scale synoptic weather systems at scales of many 1000 km. Current global and regional climate models typically operate at grid spacings on the order of 10-300 km, and are
thus unable to explicitly represent these interactions.

In conventional models, convective processes need to be treated with subgrid-scale parameterization schemes, which entail major uncertainties (Dai and Trenberth, 2004). These uncertainties not only raise concerns about the model's abilities to represent the associated feedback processes (Hohenegger et al., 2009), but also regarding uncertainties in climate change projections (Bony et al.,
35 2015).

Refining the model resolution to the kilometer scale allows omitting the parameterization of deep convection, since at this resolution the associated processes can be represented explicitly. In the last decades, this approach has successfully been followed in idealized studies (e. g. Weisman et al. (1997)) and for numerical weather prediction purposes (e. g. Benoit et al. (2002)). Convective pro-
cesses are then represented much closer to first principles and thus allow for an improved skill in quantitative precipitation forecasting (Mass et al., 2002; Richard et al., 2007) and ultimately for an improved representation of the water cycle. Recent studies have applied this approach to limited-area climate modeling: In their decade-long, regional simulations over England and the Alps, Kendon et al. (2012) and Ban et al. (2014) found significant improvements in the representation of sub-daily
precipitation events over land, in particular regarding rainfall intensity, duration, spatial extent, as well as the timing of the diurnal cycle of precipitation, especially for high precipitation percentiles (Ban et al., 2015). Following promising validation, decade-long simulations for climate scenarios have been conducted (Kendon et al., 2014; Ban et al., 2015).

While the convection-resolving approach shows very promising results, turbulent and convective
motions are only partly resolved (Wyngaard, 2004). Using numerical simulations of an idealized squall line, Bryan et al. (2003) showed that a horizontal grid spacing of 250 m and below is needed to accurately predict the details of deep convection. Associated with this limitation is a high sensitivity of condensation processes with respect to grid spacing (Bryan and Morrison, 2012). However Langhans et al. (2012) found that large-scale bulk properties of atmospheric convection, such as
moisture and temperature tendencies, converge at a grid spacing of about 2 km. Their findings indi-



cate that for real-case simulations, kilometer-scale resolution is often sufficient, provided the focus is on bulk properties and feedbacks rather than the structure of the convective clouds.

Convection-resolving simulations have proven to be very useful tools for climate simulations and numerical weather prediction (Mass et al., 2002; Lean et al., 2008; Attema et al., 2014). However the narrow grid spacing and small time steps involved represent a major challenge for current super-computers, in particular for large spatial domains and long time-scales. Therefore climate simulations with convection-resolving resolution have so far been limited to comparatively small domains (Knote et al., 2010; Kendon et al., 2012; Prein et al., 2013; Ban et al., 2014). On the global scale, this challenge is even more ambitious (Wehner et al., 2011; Palmer, 2014). Nevertheless, the exponential growth in compute power led to a number of computational breakthroughs of global simulations: Miyamoto et al. (2013) demonstrated a 12-hour-long global simulation at a grid spacing of 870 m, Miura et al. (2007) performed a week-long simulation with a horizontal grid spacing of 3.5 km, and recently Skamarock et al. (2014) performed a 20-day-long simulation with a horizontal grid spacing of 3 km. While these efforts portray the limit of what is achievable today, they also illustrate the benefits of global model formulations that overcome convection parameterization schemes.

Although designed for a wide range of potential applications, high-performance computers are not necessarily optimal for convection-resolving atmospheric models (Donofrio et al., 2009; Bauer et al., 2015). This gap can to a large degree be tied to the scaling properties of different types of algorithms: While the arithmetic intensity (ratio of floating-point operations to total data movement) increases linearly for many dense-linear-algebra operations, it remains low for the stencil computations typically found in the dynamical cores of atmospheric models (Schulthess, 2015). Consequently the stencil operations, which are commonly used in the most time-consuming parts of the code (the dynamical cores), are usually limited by the available memory bandwidth rather than the potentially available floating-point performance (Christen et al., 2008; Gysi et al., 2015).

In the last years, electrical energy constraints for supercomputers have led to heterogeneous computer architectures that involve conventional multi-core CPUs as well as attached accelerators such as graphics processing units (GPUs). For weather and climate models, GPUs are particularly interesting because their "parallelism is substantial" and because they "prioritize throughput over latency" (Owens et al., 2008). Hence they have the potential to close the performance gap needed for more extensive convection-resolving simulations. Other proposals of new computer architectures useful for weather and climate modelling are: other accelerators such as the Intel Xeon Phi architecture or FPGA-accelerators (Deest et al., 2016), custom chips (Donofrio et al., 2009) or inexact hardware (Düben et al., 2014).

Multiple efforts to port existing weather and climate codes to GPUs have been undertaken: With their pioneering work on the Weather Research and Forecast (WRF) model Michalakes and Vach-harajani (2008) demonstrated the applicability of the approach to weather and climate codes. An effort which has meanwhile been continued by Mielikainen et al. (2012) and others. Similar attempts



have been made by Shimokawabe et al. (2010) to accelerate the next version of the ASUCA produc-
tion weather model or Demeshko et al. (2013) that report on a GPU implementation of the NICAM

shallow water module. A team at the National Oceanic and Atmospheric Administration (NOAA)
have demonstrated promising performance increases for the dynamical core of the non-hydrostatic
Icosahedral Model (NIM) and are now working towards porting the NIM physics package (Hen-
derson et al., 2011; Govett et al., 2014). In the Large Eddy Simulation (LES) domain, Schalkwijk
et al. (2015) have fully ported the Dutch Atmospheric Large Eddy Simulation (DALES) model to

GPUs allowing also on-the-fly visualization. Since the introduction of general purpose GPU com-
puting, substantial speedups have been reported for dynamical cores, physics and diagnostics and
adapted techniques for inter-node communication have been outlined. However, although some of
the models have been used for real-case weather simulations (Schalkwijk et al., 2015), they usually
did not include the full suite of parameterizations or were driven by a vertical profile rather than

by time-dependent lateral boundary conditions. A proof of concept of a climate simulation using a
production quality model, computed on heterogeneous architectures, has not yet been accomplished.

In this study we demonstrate the capabilities of GPU-accelerated simulations in the area of re-
gional climate simulations, addressing week and month-long simulations on a European-scale com-
putational domain. We use a new version of the COSMO (Consortium for Small-Scale Modeling)

model enabled for GPUs (Fuhrer et al., 2014). In contrast to other projects discussed above, this
model executes all the code required for the timestepping on GPUs (dynamics, physics and diagnos-
tics), including the halo exchange at sub-domain boundaries. Execution of the entire time stepping
algorithm on the accelerators is essential to minimize expensive data movements between the CPU
and the accelerator. The code developments have recently been integrated into the operational NWP

suite at MeteoSwiss (operating with a grid spacing of 1 km) and will soon become available to the
wider COSMO community.

Using results from week-long and season-long simulations, we assess the applicability of the
convection-resolving COSMO model on continental scales. We start by presenting an outline of the
methodology (Section 2). In terms of results, we provide insights into simulated meso-scale features

such as the formation of line convection along frontal zones, the evolution of diurnal convection
over Europe during the summer season, and the role of propagating cold pools in the initiation of
convective cells (Sections 3 and 4). Afterwards we discuss the performance gained from using GPUs
for real-case simulations (Section 5) and finally conclude the study (section 6).

## 2 Methods

### 2.1 Model description

This study utilizes a refactored version of the COSMO (Consortium for Small-Scale Modeling,
v4.19) weather and climate model. The version is capable of running on heterogeneous hardware



architectures (Fuhrer et al., 2014). The COSMO model is a non-hydrostatic limited-area model that solves the fully compressible governing equations using finite difference methods on a structured grid (Steppeler et al., 2003; Förstner and Doms, 2004). It employs a split-explicit third-order Runge-Kutta discretization to integrate the variables forward in time (Wicker and Skamarock, 2002) and is discretized on a rotated latitude-longitude grid using terrain-following surfaces. The horizontal advection scheme is a fifth-order upwind scheme and in the vertical direction an implicit Crank-Nicholson scheme (Baldauf et al., 2011) is used. The multi-dimensional advection of scalar fields is implemented using the one-dimensional Bott scheme (Bott, 1989) with time splitting (Schneider and Bott, 2014). The resulting model is suitable for weather and climate simulations with spatial resolution ranging from 50 km to the kilometer-scale.

The physical parameterizations used in this study include a radiative transfer scheme based on the $\delta$-two-stream approach (Ritter and Geleyn, 1992), a single-moment bulk cloud-microphysics scheme that uses five species (cloud water, cloud ice, rain, snow, and graupel, Reinhardt and Seifert (2005)), the multilayer soil model TERRA_ML (Heise et al., 2006) with 8 active soil layers with varying layer thicknesses between 1 cm and 7.48 m and a total soil depth of 15.24 m. Furthermore a turbulent-kinetic-energy-based parametrization is used in the planetary boundary layer (PBL), and for surface transfer (Mellor and Yamada, 1982; Raschendorfer, 2001). In addition and depending upon resolution, sub-grid convection is parameterized using the Tiedtke mass-flux scheme with moisture-convergence closure (Tiedtke, 1989).

### 2.2 Enabling COSMO on heterogeneous architectures

The approach to port COSMO to heterogeneous hardware architectures with GPUs is as follows (Figure 2): The most compute intensive module (the dynamics) has been rewritten in C++, using the Stencil Loop Language (STELLA, Gysi et al. (2015)). STELLA is a domain-specific embedded language (DESL) specialized for solving stencils on structured grids. It allows aggressive low-level architecture-specific performance optimizations and the use of platform-specific programming models, while maintaining a single code syntax at higher levels of the code. During code-compilation, the stencil templates are then translated to an implementation specific to the target architecture.

For the physics, diagnostics and most of the handling of the lateral boundary conditions, a less disruptive approach has been chosen (Lapillonne and Fuhrer, 2014). Here execution and data movement is organized using OpenACC (2011) compiler directives. Directives are instructions specifying additional guidance to the pre-processor or the compiler. OpenACC directives allow a programmer to mark kernels (the body of a loop) that can be offloaded from a host CPU to an attached accelerator, and also organize their execution as well as data movement between CPU and accelerator. Although directives grant less flexibility to optimize for a specific hardware architecture (for instance changing the loop and storage order), they allow to mostly retain the existing FORTRAN code, and make it possible to port large portions of code quite fast.



In large simulations, the computational domain is usually split into smaller subdomains (domain
decomposition). The data exchange required at the sub-domain boundaries (i.e. halo exchange) is
handled using a re-usable communication framework. It guarantees performance portability across
different high-performance computing architectures by leveraging the capabilities of the Generic
Communication Library (Bianco, 2012). Similar to STELLA, the GCL abstracts the complicated
pathways that move data through heterogeneous machines. With this approach, the time stepping
runs entirely on accelerators. This property is fundamental to a fast performance, as the memory
footprint of the prognostic variables in the simulations to be presented amounts to 96 Bytes per grid
point. Moving such a large footprint each time step (between CPU and GPU), while only perform-
ing a comparatively small amount of floating-point operations per transfer, would be prohibitively
expensive. In other words, the memory transfer GPU and CPU is simply too slow to make back and
forth transfers worthwhile at each time step.

The modules written in C++ and FORTRAN are integrated by a C++ interface which provides
FORTRAN bindings. For a detailed outline of the software engineering approach of the COMSO-
GPU port please see Fuhrer et al. (2014).

### 2.3 Model setup

The model is used in two configurations (Figure 1): The first configuration uses parametrized shal-
low and deep convection at a grid spacing of 12 km and a domain size of 355x355x60 grid points
(CTRL12). The second configuration has a convection-resolving horizontal grid spacing of 2.2 km
and 1536x1536x60 grid points (CTRL2). In this configuration, the deep-convection parameterization
is switched off, but the shallow-convection scheme remains active. Here, the parameterized fraction
of (shallow) convective clouds is non-precipitating and has a maximum vertical extent of 250 hPa,
while deep convection is treated explicitly. Following the recommendations by Baldauf et al. (2011),
in CTRL2 the Mellor-Yamada asymptotic length scale in the PBL parameterization is reduced by a
factor 2.5 to calibrate the triggering of convection. In both models, the vertical direction is discretized
using 60 stretched model levels from the surface to the model top at 23.5 km. The respective layer
thickness widens from 20 m at the surface to 1.2 km near the model top. Aside from the domain size,
we generally follow the setup defined in Ban et al. (2014).

The CTRL12 domain spans about 4300x4000 km and thereby covers most of continental Eu-
rope including the Mediterranean. The domain for the CTRL2 simulation is approximately 500 km
smaller than the CTRL12 domain (on each side), but still covers most of Western and Central Europe
(Figure 1). The necessary initial and boundary conditions for the CTRL12 simulation are derived
from the European Centre for Medium-Range Weather Forecasts (ECMWF) ERA-Interim reanaly-
sis (Dee and Uppala, 2011) and are updated every 6 h. Using two-step one-way-nesting, the results
from the CTRL12 simulation are subsequently used to derive boundary conditions for CTRL2 at an
hourly interval. The analysis domain excludes grid columns close to or within the relaxation zone





(50 km distance to the CTRL2 boundary). Additionally a simulation with a grid spacing of 50 km has been performed (CTRL50). Apart from the horizontal resolution and the associated time step, it has the same setup as CTRL12. This simulation portrays the current generation of high-resolution global climate models and its results are used for illustrative purposes in figure 6 and in the supplementary material (Figure S1).

## 2.4 Numerical experiments


Here we present results for two model integrations: a week-long winter case with strong synoptic forcing, and a seasonal integration of a summer case that is characterized by a rather weak synoptic forcing. For the winter case, the model chain has been initialized on 16 January 2007 00 UTC and integrated for seven days until 23 January 2007 00 UTC. For the summer case, the 2-km simulation
is initialized on 1 Mai 2006 and integrated until the end of August.

To provide adequately spun-up soil moisture fields for the summer 2006 simulation, the soil layers in CTRL12 have been initialized on 1 May 2001 based on the soil-moisture fields from the CCLM EURO-CORDEX simulation (Kotlarski et al., 2014), and thereafter integrated for 5 years. Subsequently CTRL2 has been initialized on 1 Mai 2006 and integrated for four months. The analysis
period for this simulation has been defined as 1 June 2006 to 31 August 2006 (JJA) leaving one month of CTRL2 integration for spin-up.

For the summer 2006 simulation we also tested a parameter calibration based on the findings of Bellprat et al. (2016). They demonstrated a "pronounced reduction of the summer warm bias" by introducing objectively calibrated values of 8 model parameters. Application of their calibration to
the current setup resulted in a reduction (domain-mean, all land points) of the warm bias by about 0.7°C (see supplementary material Figure S3) and an increase of the seasonal mean-precipitation by about 0.2 mm/day (see supplementary material Figure S2).

For the winter case, the model chain has been initialized on 16 January 2007 and integrated for seven days. As winter simulations are less sensitive to soil conditions, the initial soil and snow data
was directly taken from ERA-Interim. The initial 36 h of the simulation are considered spin-up and are not analyzed in any detail.

### 2.4.1 Pseudo-synthetic visualization of Clouds

An attractive method to visualize clouds is to compute synthetic brightness temperatures during model integration through the use of a forward radiative transfer model. In the COSMO model, the
RTTOV satellite simulator is being used for this task (Keil and Reinhardt, 2006). Unfortunately this functionality is not yet available in the GPU version used in this study. To circumvent this limitation, and nevertheless provide 2D cloud visualizations, we combine bulk-diagnostic cloud fractions into a pseudo temperature ($B$). Cloud fractions are a diagnostic that represent, how the radiation scheme interacts with clouds. Thereby the 3D fields are aggregated onto three two-dimensional cloud-fraction



fields during model integration (low ($f_{lc}$), mid $f_{mc}$ and high clouds $f_{hc}$). Essentially the cloud fraction diagnostic provides a summary measure of the cloud-covered fraction of a grid cell.

The conversion of the three cloud fractions into one single brightness is accomplished by using four calibrated parameters $m$ as follows: In a first step $B_{srf}$ is assigned a surface brightness value based on the underlying land-cover ($m_{srf}$ over land and 0 over sea):

$$B_{srf} = \begin{cases} m_{srf}, & \text{if land point} \\ 0, & \text{if sea point} \end{cases} \tag{1}$$


In the next step, the pseudo brightness temperatures of low ($B_{lc}$), mid ($B_{mc}$) and high levels ($B_{mc}$) are multiplied by a parameter ($m_{lc}$, $m_{mc}$ and $m_{hc}$) and successively stacked on each other, while also taking into account the clouds on lower layers:

$$B_{lc} = f_{lc} * (m_{lc} - B_{srf}) + B_{srf} \tag{2}$$

$$B_{mc} = f_{mc} * (m_{mc} - B_{lc}) + B_{lc} \tag{3}$$

$$B_{hc} = f_{hc} * (m_{hc} - B_{mc}) + B_{mc} \tag{4}$$

The final quantity, i.e. $B_{hc}$, is meant to mimic a brightness that can qualitatively be compared with satellite images. To this end, the parameters $m$ are calibrated as follows: $m_{srf} = 0.15 < m_{lc} = 0.2 <$
$m_{mc} = 0.3 < m_{hc} = 1$. A visual comparison of the pseudo-synthetic satellite images and synthetic RTTOV images can be found in the supplementary material (Figure S1).

## 3  Meso-scale features in a week-long European-scale convection-resolving simulation of winter storm Kyrill

In January 2007 the devastating winter-storm Kyrill passed over northern Europe, tremendously
affecting infrastructure and sadly also human lives. While often referred to as "Kyrill", the storm considered actually was a sequence of two extratropical cyclones. Based on a backward trajectory analysis Fink et al. (2009) outline that the first storm (Kyrill I) emerged from a "cold front located underneath the eastward side of an upper-level, long-wave trough over North-Eastern Arkansas (USA)" on 14 January 2007. Traversing the North Atlantic, the storm underwent rapid cyclogenesis while
crossing the jet-stream from the warm to the cold-side. On 18 January at 00 UTC a second storm (Kyrill II) formed on the occluded front of Kyrill I northwest of Ireland.

In their modeling study, Ludwig et al. (2015) describe the dynamical forcing leading to the Kyrill II cyclogenesis as an interaction between frontolytic strain acting on a low-level potential vorticity filament of the occluded front of Kyrill I, and a developing upper-level short-wave trough. In a
series of sensitivity experiments, they also determined that the diabatic heating processes between



800 and 500 hPa posed an additional crucial ingredient for the cyclogenesis for the development of Kyrill II. Using a convection-parameterizing 25 km COSMO setup, they show that latent heating accelerated cyclogenesis, and also increased the core pressure drop by 10-15 hPa. These studies show that close interaction between upper-level divergence and low-level baroclinicity, but also that

diabatic processes were key in its development. Therefore this case poses an interesting challenge to test our models. A useful feature of this episode was the presence of "a remarkable pressure gradient of more than 70 hPa [..] between [...] the North Sea and [...] the Iberian Peninsula" (Fink et al., 2009), which should confine the internal variability of the large-scale circulation.

### 3.1 Results

The overall surface development of Kyrill II on the 18th of January is as follows (Figure 3): The Kyrill II storm develops along a pronounced baroclinic zone around 00 UTC northwest of Ireland (see arrow in Figure 3 at 06 UTC), then it rapidly propagates over the UK, intensifies in the North Sea (around 12 UTC) before reaching the continent (around 18 UTC). For the time period considered here, the overall solutions of CTRL12 and CTRL2 agree well.

Along the prominent pressure gradient between the Alpine ridge and Scandinavia, the isobars follow a very similar path across Europe. The surface pressure and 2m-temperature fields of CTRL12, and even more noticeable of CTRL2, exhibit small-scale wave patterns, in particular in the vicinity of topographic areas. We interpret these features as gravity-wave signals and small-scale temperature variations associated with the underlying topography. The resolution of ERA-Interim is too coarse

to represent these features. The biggest difference in sea-level pressure and surface temperature can be seen along mountainous areas. Note that figure 3 displays the variables in their native resolution without any smoothing and hence some additional artifacts may be present due to reducing surface-pressure to mean-sea-level pressure.

Shortly after cyclogenesis, the core pressure in the reanalysis is lower, however the horizontal

temperature gradient along the warm front is already steeper in the simulations. Six hours later, a similar situation is evident also on the cold front. While the simulations expose a steep horizontal temperature gradient, it is less clear-cut for ERA-Interim. During cyclogenesis of Kyrill II and while passing the British Isles, a small low-pressure system is present in the west of the Norwegian coast. Later on, at 12 UTC, the simulations expose two separate systems, but in ERA-Interim the 974 hPa

contour encloses both systems. Consequently the spatial extent of Kyrill II appears to be smaller in CTR12 and also in CTRL2. When the storm sweeps over Denmark, at 18 UTC, intensity and location of the storm again agree well, but warm and cold sector are still more defined in the simulations than in ERA-Interim. For a visualization of the simulated storm with high temporal resolution see the following video: Leutwyler et al. (2015a)

The situation at upper levels (Figure 4) corroborates that the simulated geopotential heights qualitatively agree with the driving ERA-Interim data. However the temperature gradients near the cy-





clone core, on the 850 hPa and 500 hPa pressure levels, are much more pronounced, with colder temperatures to the north of the cyclone. The most prominent difference on the 200 hPa level is the slightly higher temperature maximum over Denmark.

Figure 5 compares the cyclone's surface core pressure of our simulations (CTRL12 and CTRL2) with two simulations of Ludwig et al. (2015) (LW25 and LW7), and with the ERA-Interim reanalysis. It should be noted that the observational reference is rather weak, as this was a rapidly developing small-scale cyclone. In comparison to ERA-interim, all four simulations exhibit a higher initial surface pressure (lowest local minimum) at the time of cyclogenesis and maintain it until the inten-

sification phase starts around 10 UTC (Figure 5). Then the simulations exhibit a core pressure drop by about 9 to 12 hPa in 7 h, significantly below the ERA-Interim estimate. While this behavior is rather distinct from the evolution in ERA, the four simulations qualitatively agree. However, LW25 and LW7 show a recovery towards the ERA-Interim values when Kyrill II makes landfall, while in our simulations core pressures below 960 hPa prevail until the storm exits the domain. To further

investigate these differences, we conducted an additional simulation with the CTRL12 configuration, but with the same domain setup as LW25. In contrast this simulation followed the core pressure recovery of LW25.

As shown by Ludwig et al. (2015), the case is strongly sensitive with respect to latent heating. Weaker latent heating rates reduce the core pressure drop and delay cyclogenesis. Since we generally

observe more precipitation in CTRL2 than in CTRL12 (not shown) we consequently also expect deeper core pressures, which is consistent with figure 5.

As outlined above, the two COSMO model simulations find comparable synoptic-scale solutions. This is also true for precipitation and synoptic-scale clouds. On 17 January 2007 12 UTC, a large low-pressure system is located north of the British Isles with an attached, elongated cold front (Fig-

ure 6, top panels). As expected the CTRL12 and CTRL2 simulations reveal an increasingly higher level of detail than CTRL50, stronger gradients and smaller precipitating regions of higher intensity, while the meso-scale spatial structure of precipitation and clouds are rather consistent. On the level of individual precipitating systems on the other hand, there are pronounced differences between CTRL12 and CTRL2. First, the cold front is associated with a narrow band of convective clouds.

This band is well captured also by CTRL12, but substantially narrower with resolved convection. Second, the cold frontal passage yields heavy convective activity in CTRL2 which also occurs in CTRL12, but less pronounced. During the passage of Kyrill II over central Europe on 18 January 2007 18 UTC, the differences are even more pronounced (Figure 6, bottom panels). While CTRL50 shows a closed cloud cover with light precipitation below 10 mm/h, the horizontal variability in

CTRL12 is already larger and the cloud cover is split into smaller systems. In CTRL2 the horizontal variability is again increased and many small convective systems and cells can be found. Some signatures of the small scale systems can also be found in the geopotential height field. Particularly noticeable are the changes for the region with a precipitation intensity above 5 mm/h found



in CTRL50 (red area in the bottom-left panel in figure 6). In CTRL12 this area is split into succes-
sive precipitation bands with maxima up to 20 mm/h. Besides the sharpened and even more intense
rain bands (up to 50mm/h), CTRL2 additionally features small embedded convection located in the
vicinity and along the cold front. We expect these differences in location and intensity, due to the
ability of CTRL2 to explicitly resolve the underlying dynamical systems.

Next we discuss the model representation of a low-level, meso-scale eddy, which is located behind
the cold front of the displayed low-pressure system on 17 January 2007 12 UTC (orange box in fig-
ure 6). A zoom of this area is shown in figure 7. These meso-scale eddies or polar lows typically fea-
ture strong convective activity and therefore pose an interesting challenge for convection-resolving
models. In both simulations an eddy can be inferred from the bend in the 850 hPa geopotential height
contour (Figure 7, top). However, while the geopotential height contours compare rather well, the
associated precipitation pattern, does not exhibit much similarity. The precipitation maximum in the
tail of the eddy (13° W, 55° W) can be found in both simulations, but the higher resolution enables
a more coherent organization of convective cells. In particular downstream of the vortex, CTRL12
produces many isolated precipitating grid points, while CTRL2 shows well-developed signs of or-
ganization and wrap up. The scale of the CTRL2 simulated features amounts to typically 4-7 grid
points. CTRL12 does not seem to exhibit this organization.

The explicit representation in CTRL2 is also evident in the distribution of hydrometeors (rain,
snow and graupel). The vertically integrated distribution of hydrometeor mass (Figure 7, middle
panels) is spatially more confined in CTRL2 and thus testifies the role of significant updrafts, while
in CTRL12, significant hydrometeor loads can only be identified at the precipitation maximum,
discussed above.

The temperature fields at 850 hPa also reveal a consistent picture (Figure 7, bottom). While
CTRL2 exhibits a distinct wrap-up structure, an eddy-like pattern can hardly be identified in CTRL12.
Additionally, the diagrams reveal small-scale superimposed anomalies stemming from diabatic heat-
ing. In CTRL2 they are arranged in a circular fashion around the eddy core, while in CTR12 they
are much less organized.

It should be stressed that a thorough validation is here not attempted for several reasons. First, as
can be deduced from alternate simulations that were initialized 6, 12, 18 and 24 h hours earlier (not
shown), the predictability of this particular small-scale vortex is very small. Second, as the current
version of COSMO-GPU lacks a GPU-enabled version of the RTTOV (Keil and Reinhardt, 2006),
a thorough validation with satellite pictures would be dubious. However, the preference of CTRL2
to form small-scale vortex-like features (that wrap up) is very common in the simulation discussed.
Consistent with these results McInnes et al. (2011) found, in their model study on polar lows, that
decreasing the grid spacing to the kilometer scale improves the representation of meso-scale ed-
dies. Among other things, they found a more realistic wind field and a more realistic distribution of
convection.





## 4 A seasonal simulation of the summer 2006

Persistent large-scale anticyclonic flow was the dominant circulation pattern in Europe during the summer season 2006. Strong diurnal convection and thunderstorms could be observed, and July 2006 was the hottest month measured in Europe to this date (Rebetez et al., 2009). Not only for that
reason, this month has been the subject of some previous studies (e. g. Hohenegger et al. (2009)). During such anticyclonic episodes, the lateral boundary conditions typically exert less control on the atmospheric circulation, and local drivers become more important. In these situations, RCMs can develop flow patterns which substantially deviate from the driving model. In order to test the model also under these conditions, a three-months-long simulation of this episode was conducted.

### 4.1 Results

### 4.1.1 Seasonal statistics

Over-prediction of summer temperature is a long standing issue for COSMO-CLM and other RCMs, in convection-parameterizing (Kotlarski et al., 2014) as well as in convection-resolving setups (Ban et al., 2014). Validation of the CTRL2 average summer 2m temperature (JJA), using the E-OBS
dataset as observational reference (Haylock et al., 2008), shows that this behavior is still an issue (Figure 8, left-hand panel). After accounting for differences in topography and spatial resolution (height correction with a lapse-rate of 0.65 K/100m), the resulting domain-mean warm bias amounts to about 1.4 °C, with the largest biases in Northern Africa and Eastern Europe. The large warm biases in Northern Africa and Eastern Europe are known RCM biases and not only related to model
biases, but also to data sparsity in the verification data sets (Kotlarski et al., 2014; Panitz et al., 2014, and references therein).

The spatial distribution of precipitation is well captured (Figure 8). Simulated precipitation over elevated topography is much larger than the observations, but this is, at least partly, related to the sparse observational network used to create the E-OBS precipitation dataset (Hofstra et al., 2009;
Isotta et al., 2015). This observational bias is also attenuated by the biased distribution of rain gauges, which are predominantly located in valleys where precipitation is typically much smaller than at mountain peaks. The increase precipitation magnitude is also reflected in the domain average land precipitation which is 2.1 mm/day in CTRL2 and 1.8 mm/day for E-OBS.

While the spatial distribution of precipitation agrees well between CTRL12 and CTRL2, their
behavior on the sub-daily timescale is fundamentally different (Figure 9): The different timing of the diurnal cycle (left-hand panel) is remarkable. While the convection-parameterizing CTRL12 simulation is already at its peak around noon, the convection in CTRL2 is still building up and peaks only later in the afternoon. Furthermore the mean daily maximum precipitation is higher in CTRL2 (right-hand panel) and also produces larger hourly precipitation maxima (middle panel).
It has previously been shown for smaller domains (Kendon et al., 2012; Ban et al., 2014) that the



behavior of the convection-resolving model fits observation much better. Our results are qualitatively consistent with these studies, although the differences in daily precipitation statistics are larger for our simulation. Note, however, that Ban et al. (2014) considered the statistics from 10 summers, while here only one summer season is considered. A more detailed validation of precipitation will
be conducted in a subsequent study using a 10-year-logn climate simulation.

A snapshot on a typical summer day at noon illustrates how the different precipitation event distributions come about (Figure 10 and Leutwyler et al. (2015b)). In the cloud field of CTRL2, the convective cells are visible as high-reaching, initially circular, cloud features. In CIRL12, on the other hand, the convection-parameterization schemes adjusts the vertical stability of the atmosphere
before grid-scale convective motions can develop, and consequentially convective cells are absent. In CTRL12 precipitation is characterized by widespread light rain below 5 mm/h with occasional patches exceeding that threshold. In contrast CTRL2 shows smaller isolated cells and convective cores with an intensity above 10 mm/h. This bahavior of CTRL2 consequentially leads to higher hourly peak amounts, as noted in figure 9.

### 4.1.2   Propagating cold pools

Cold-pools are formed by cold negatively-buoyant air, stemming from evaporation of falling hydrometeors. The associated downdrafts penetrate into the planetary boundary layer and locally enhance the variability of the moisture, temperature and wind fields. Their role in the initiation and organization of deep convection over land has been studied using radar observations Lothon et al.
(2011); Dione et al. (2014) as well as Large Eddy Simulations (LES) (Grabowski et al., 2006; Khairoutdinov and Randall, 2006; Boing et al., 2012; Schlemmer and Hohenegger, 2014). While idealized simulations and flat semi-arid regions provide an ideal environment for the formation of large cold pools, they are less pronounced in more heterogeneous areas, such as in continental Europe. How these processes are represented in the convection-resolving simulation is illustrated in the
following video: Leutwyler et al. (2016).

The use of high-resolution models covering large domains provides a tool to study cold pools in heterogeneous areas, and we here focus on the subdomain indicated by the red box in figure 10. At 12 UTC a few small precipitating cells are present. An hour later, at 13 UTC, the cells have grown in size and a number of them exhibit signatures typical for cold pools (Figure 11). For instance in
900 hPa vertical wind field, circular downdrafts, surrounded by a ring of updrafts, appear below precipitating convective cells. They overlap with distinct local temperature minima. In the subsequent snapshots, at 13:30 and 14 UTC, the cold pools grow in size and some of the cells start to develop strong dry downdrafts. At the same time, the anvil clouds are expanding.

In order to assess whether new cells are triggered along propagating cold pools, a subjective tracking of cold-pool signatures is applied. To this end, the convective cells, visible in the snapshots taken
at 13:30 and 14 UTC, which are not present in the previous snapshot, have been marked with a black



circle in the left-most panels. Subsequently the same locations have been marked in the respective vertical wind panel of the previous snapshot. It can be seen that a number of black circles are co-located with the moving edges of the cold pools, but also with convergence lines and topographic

features. The co-location of new cells and leading cold-pool edges confirms that propagating cold pools are relevant for the initiation of new convective cells in simulations. The results are qualitatively consistent with LES results found in idealized studies (Schlemmer and Hohenegger, 2014). As expected, no corresponding signatures have been found in the CTRL12 simulation (not shown).

## 5    Computational requirements

What are the computational requirements to perform a convection-resolving simulation on the European scale? There are many elements involved in designing such an experiment. Here we restrict the analysis to two key performance metrics: First, on the achievable time to solution for a fixed simulation domain size, while increasing the computational resources (strong scaling). Second, on the time to solution achievable when the simulation domain is increased proportionally with the computa-

tional resources (weak scaling). Furthermore we assess the performance gain from using GPUs with respect to conducting simulations on multi-core hardware. Here we test a single code version that is able to run on different hardware architectures (with and without GPUs). In contrast to Fuhrer et al. (2014), we use a real-case, climate configuration close to what has been described in section 2.4, also accounting for both input and output.

On a distributed memory system, the problem considered here needs to be split into smaller chunks and hence messages have to be communicated across the network. In COSMO, this is achieved by decomposing along the horizontal dimensions. This domain decomposition yields a communication pattern where four messages are transferred to the four neighboring compute nodes: north, south, east and west. When a computation is distributed onto an increasing number of nodes, the ratio between

the amount of computation on a node and the amount of information exchange with neighboring nodes decreases. In a simple performance model, the speedup from parallelization will saturate towards a theoretical value and is proportional to the square root of the number of sub-domains and a machine constant (Wehner et al., 2011). On most CPU-based hardware, this limitation creates a lower bound on the time to solution, which can be achieved for strong scaling. On heterogeneous

hardware equipped with GPU's, the end of strong scalability may be reached earlier (Fuhrer et al., 2014).

    The full strong-scaling experiment corresponds to a 24 h simulation on a domain of 1536x1536x60 grid points. Input for this simulation consists of the lateral boundary conditions at hourly resolution, amounting to about 120 GB for the whole simulation . Additionally an output workload consisting

of about 6 GB is written to the file system. All performance results have been obtained on a heterogeneous Cray XC30 system, located at the Swiss National Supercomputing Centre (CSCS) in



Lugano, Switzerland (Piz Daint). This machine consists of a heterogeneous node architecture with an eight-core Intel Xeon E5-2670 CPU and an NVIDIA Tesla K20X GPU per node, and Cray's Aries interconnect using a three-level dragonfly topology to connect the compute nodes.

There are several options to compare heterogeneous and non-heterogeneous node architectures. A pragmatic way to normalize performance metrics is to define them as per socket. A socket is the electrical component that provides the connection between the circuit board and the chip sitting on top of it. Another metric would be node-to-node comparison, assuming that a node can either consist of one CPU and a GPU, or of two CPUs. We believe that the per-socket performance metric is more

useful than node-to-node comparisons, since nowadays fat-nodes are commercially available. These nodes are equipped with multiple GPUs but only a single CPU, making node-to-node comparison less meaningful.

### 5.1 Scaling

In the first experiment (strong scaling), the time to solution for a 24h simulation on 1536x1536

grid columns, distributed among an increasing number of sockets, is measured (Figure 12, left-hand panel). The time to solution for execution on the CPU decreases approximately linearly up to 900 sockets which corresponds to 2620 grid columns per CPU socket and 328 grid columns per MPI task. Towards the end of the curve at 128 grid columns per MPI task, inter-node communication starts to limit the speedup the additional CPU-sockets provide. Execution on the GPU shows saturation

already at 256 sockets, which corresponds to 64x64 grid columns per socket. Consequently when using GPUs, a larger number of grid points per socket is needed to efficiently utilize the hardware. A similar behavior is found by Fuhrer et al. (2014) in their experiments using the same model, but with periodic boundary conditions and without I/O. They found a linear scaling behavior for experiments with more than 128x128 grid columns per socket, but also early saturation, as the workload per

socket decreases. In this study we reach the upper memory limit of the sockets earlier and therefore are not able to reproduce the linear scaling regime they find. We nevertheless found a significant speedup when using the GPU. For our reference setup with 128x128 grid columns per socket (as used in sections 3 and 4) we measured a speedup of about a factor 3.6. For a similar time to solution with the conventional CPU-based multi-core architecture, 5 times more sockets would be needed.

In the second experiment (weak scaling), the number of grid columns per socket is kept constant at 128x128, while proportionally increasing the domain size and the number of sockets (Figure 12, right-hand panel). Execution on the CPU and the GPU both show only a slight upwards trend for the time to solution. Since the performance for the physics and dynamics modules as well as data copy (to and from the GPU memory) mostly stays constant, the trend is likely related to the increase

in the amount of data written to disk as the domain size increases. In the GPU-version used in this study Input/Output is done in a synchronous manner, meaning that the model integration is stopped during file-system access. This limitation has already been addressed in a later version of the





COSMO model, and in the future we will use asynchronous I/O. For now the time-compression ratio
(simulation period divided by time to solution) for our reference setup (1536x1536 grid columns
distributed onto 144 nodes) is 1:70 with model Output and 1:90 without.

## 5.2 Assessment

Based on these benchmarks we now assess the feasibility of a large convection-resolving climate
modeling experiment, using the same domain as EURO-CORDEX (Jacob et al., 2014). This model
inter-comparison and climate projection effort consist of contributions from multiple RCMs. It in-
volves a 20 year evaluation experiment driven by reanalysis, as well as a 55 years control ex-
periment and two 94 year-long transient scenario simulations driven by a Global Climate Model
(GCM). For a simulation with a 2.2 km grid spacing, the EURO-CORDEX domain consists of
roughly 2300x2300x60 grid points. On this domain, the COSMO GPU version would yield a time-
compression ratio of about 1:60 when executed on 324 Nvidia K20x sockets and about 1:20 when
Intel E5-2670 sockets are used. Projecting the time-compression ratios on the EURO-CORDEX
experiment yields a time to solution of about 4 months for the 20 year-long evaluation period, 11
months for the 55 year-long control experiment and 1.6 years for each of the 94 years-long transient
scenario simulations.

For climate simulations, the required operational time-compression ratios are more relaxed than
in operational weather forecasting. While the workflow for weather forecasting typically imposes
strict time-to-solution constraints, the required throughput for climate simulations is governed by
more practical matters such as the duration of a project. In this regard, imposing a maximum time-
to-solution constraint of 3 months would entail a time-compression ratio of 1:500 to accomplish
a transient convection-resolving climate experiment. For comparison, in their assessment of global
convection-resolving models, Wehner et al. (2011) impose a much tougher constraint of 1:1000.
However, their CMIP5-type experiments (Taylor et al., 2012) also involve a large simulation ensem-
ble and hundreds of years of simulation for ocean spin-up. Given the 1:500 constraint, our model
would require an additional speedup of about a factor 8-10 to meet the required time-compression
constraints for an extensive CORDEX-type experiment. These results indicate that, for the COMSO
model, using GPU accelerators, permits to perform multi-year, convection-resolving simulations on
large, continental-scale domains. However, for century-long simulations at the current resolution, or
for simulations with finer grid spacing (and decreased time step), further performance improvements
are needed. We suggest that future work should focus on trying to push the strong scalability further
and thus to reduce the time required to update a gridpoint by one timestep, for example by exposing
more parallelism (in the vertical, across modules in the code, by asynchronous execution of indepen-
dent work, etc.) Another interesting application in the RCM domain would be to increase the time
step (at coarser grid spacing) and downscale a large number of GCM scenario realizations. At the
12.5 km grid spacing, used in the EURO-CORDEX EUR-11 simulations (Jacob et al., 2014), exe-



cution of the COMSO-GPU version on 10 Piz Daint nodes would fulfill the 1:500 time-compression
criterion and thus should enable a more extensive set of transient scenario simulations.

What does this mean for GCMs? For an idealized setup, Fuhrer et al. (2014) demonstrated perfect weak scaling behavior of the COSMO-GPU version up to 2000 nodes. So let us assume perfect weak-scaling (essentially neglecting Input/Output) and availability of the entire Piz Daint supercomputer (5272 hybrid Cray XC30 nodes). At a grid spacing of 2.2 km, it should technically be
possible to extend the domain size to cover about half of Earth's surface, while still retaining the same time to solution as demonstrated above. At a grid spacing of 2.8 km the whole planet could be covered. Please note that in GCMs typically other numerical methods than those used in COSMO (see section 2), with different scaling properties, are favored. Nevertheless, the analysis indicates that global convection-resolving simulations are feasible today on large dedicated supercomputing
systems, provided a code version is available that scales similar as the regional COSMO model used in the current study.

## 6   Conclusions and Outlook

The applicability of the convection-resolving climate simulation approach has been demonstrated on European scales with a new version of the COSMO weather and climate model, capable of running
on GPUs. A validation of a week-long simulation of the winter storm Kyrill showed a high level of agreement between a convection-parameterizing and a convection-resolving simulation for the development of temperature, geopotential height, core-pressure and the large scale patterns of clouds and precipitation. While in the week-long winter case rather similar solutions were found for both models, a three-month-long simulation of the summer season 2006 revealed a different character
of summer convection for the simulation with resolved convection. The precipitation field of the convection-resolving 2km simulation, showed high precipitation rates over small areas, while the convection-parameterizing, 12km simulation showed low precipitation rates over larger areas. A comparison of the diurnal cycle of precipitation of the convection-parameterizing and the convection resolving simulations showed that the results found in previous studies also apply to European-
scale domains. That is, convection-parameterizing simulations have a distorted diurnal cycle with a precipitation peak around noon, while the convection in the 2-km simulation peaks only in the late afternoon.

The simulations also demonstrated how the approach allows for the representation of interactions between synoptic-scale and meso-scale atmospheric circulations at scales ranging from 1000 to 10
km. Three examples of such interactions were discussed: Convection embedded in fronts, small eddies over the ocean in winter, and the formation and organization of propagating cold pools over land. Please note that, although we highlighted individual meso-scale systems, we did not verify their realism, structure and evolution in detail. These results illustrate some advantages of formulating




weather and climate models closer to physical first principles and portrays the benefits of using
continental-scale domains for convection-resolving models.

A substantial speedup of the simulations was realized when executing COSMO on GPU accelerators. However, at least for our hardware environment, 128x128x60 grid points per GPU were required to have sufficient work available. With the current code and the current generation of GPUs, century-long convection-resolving simulations (or further increasing the resolution) will still be chal-
lenging. For now, the GPU version of COSMO enabled us to increase the size of the computational domains to decade-long simulations, or to perform experiments with a large number of ensemble members at lower resolution.

Our next simulation target is a 10-year-long reanalysis-driven simulation covering the time period 1999-2008, using the same set up as in the current study. This simulation has already been completed
and will be analyzed in a subsequent study. It allows for a more robust validation with observational datasets. Together these simulations will serve as a proof of concept and demonstrate that convection-resolving climate simulations are feasible on continental scales.

### 6.1 Code and data availability

The particular version of the COSMO model used in this study is only a prototype and will be discon-
tinued soon. However, the code developments are currently in the process of being re-integrated into the mainline COSMO version and will soon be available to the wider research community. COSMO itself may be used for operational and for research applications by the members of the consortium. Moreover, within a license agreement, the COSMO model may be used for operational and research applications by other national (hydro-) meteorological services, universities and research institutes.
The model output encompasses 15 TBytes and is available upon request.

*Acknowledgements.* This work was supported by the Swiss National Science Foundation under Sinergia grant CRSII2_154486/1 and by a grant from the Swiss National Supercomputing Centre (CSCS). We would like to acknowledge the many contributors to the new version of COSMO used in this study. Among others: Andrea Arteaga, Mauro Bianco, Isabelle Bey, Ben Cumming, Tiziano Diamanti, Tobias Gysi, Peter Messmer, Carlos
Osuna, Anne Roches, Stefan Rüdisühli, and Thomas C. Schulthess. Also, the authors would like to acknowledge the Center for Climate Systems Modeling (C2SM) and the Federal Office of Meteorology and Climatology MeteoSwiss for their support. We thank Patrick Ludwig for providing his simulation setup and the core-pressures of the Kyrill winter storm diagnosed in his study. Furthermore we like to acknowledge the E-OBS data set from the EU-FP6 project ENSEMBLES (http://ensembles-eu.metoffice.com) and the data providers in the ECA&D
project (http://www.ecad.eu).



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





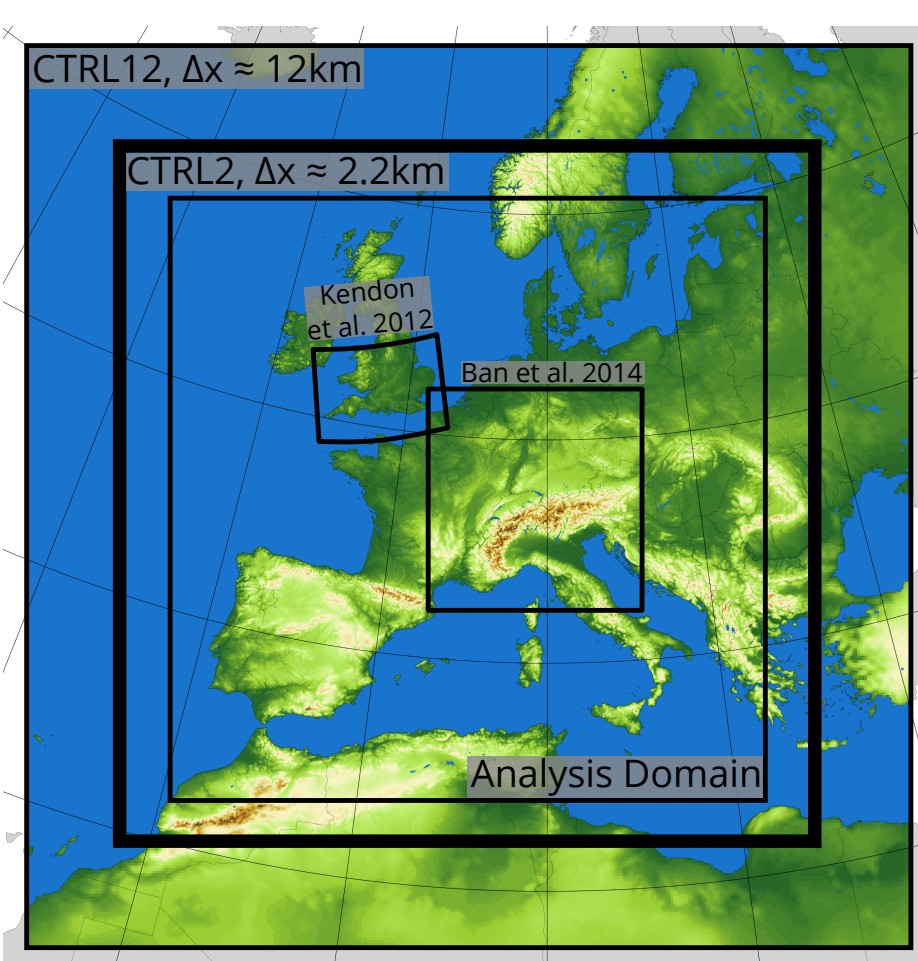

**Figure 1.** Integration domains and model topography [m]. The outermost black box show the domain of the convection-parameterizing simulation with grid spacing of 12 km, and the bolder inner box that of the convection-resolving simulation with 2.2 km grid spacing. The sub-domain used in the analysis is indicated. The two smaller black boxes indicate the domains used in two state-of-the-art convection-resolving climate simulations over the UK and the Greater Alpine Region.





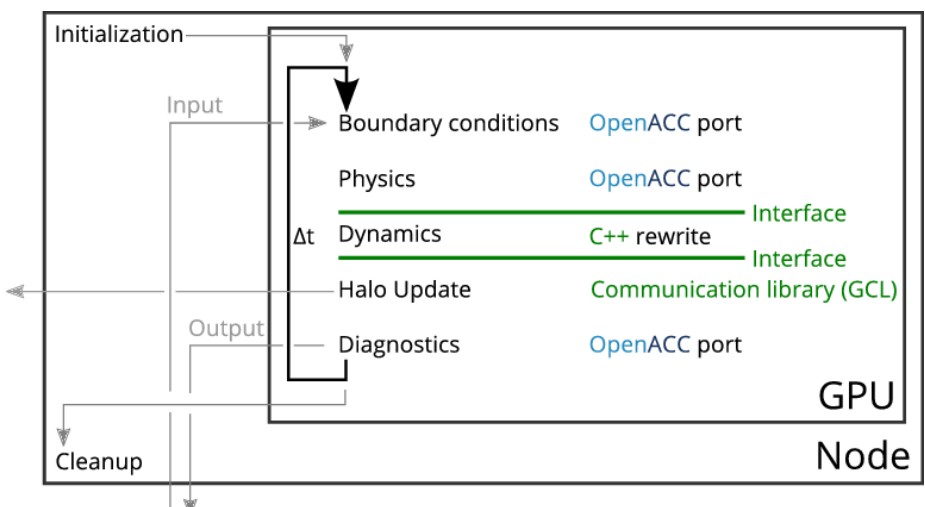

**Figure 2.** Workflow of the COSMO model on GPUs. Boundary conditions, physics, diagnostics and I/O have been ported using OpenACC (blue). Dynamics and Halo-updates have been rewritten in C++ (green).




**Figure 3.** Snapshots of the Kyrill II winter storm in ERA-Interim (left column), CTRL12 (middle column) and CTRL2 (right column) in their native resolution. The shading denotes raw 2m Temperature [°C] and the black contours mean sea-level pressure [hPa]. The contour-level spacing is 4 hPa.





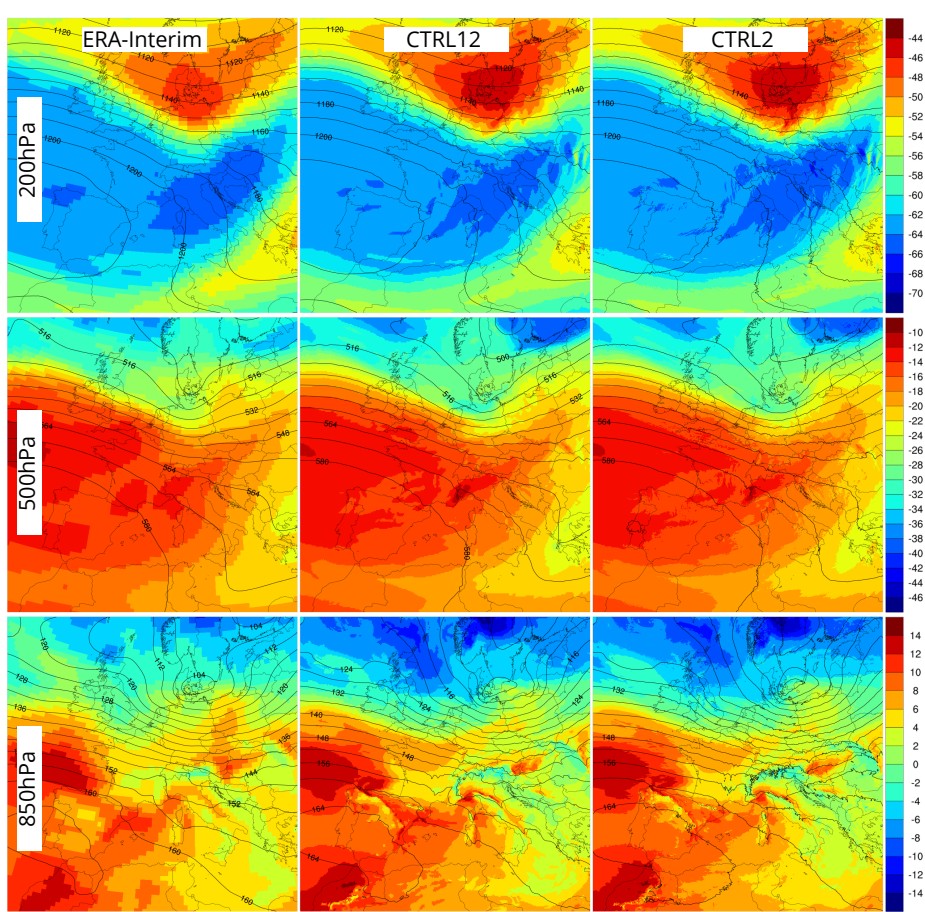

**Figure 4.** Snapshots of the Kyrill II winter storm in ERA-Interim (left column), CTRL12 (middle column) and CTRL2 (right column) at three pressure levels on 2007-01-18 18 UTC. The shading denotes Temperature [°C] and the black contours geopotential height [gpdm] with a contour-level spacing of 4 gpdm.



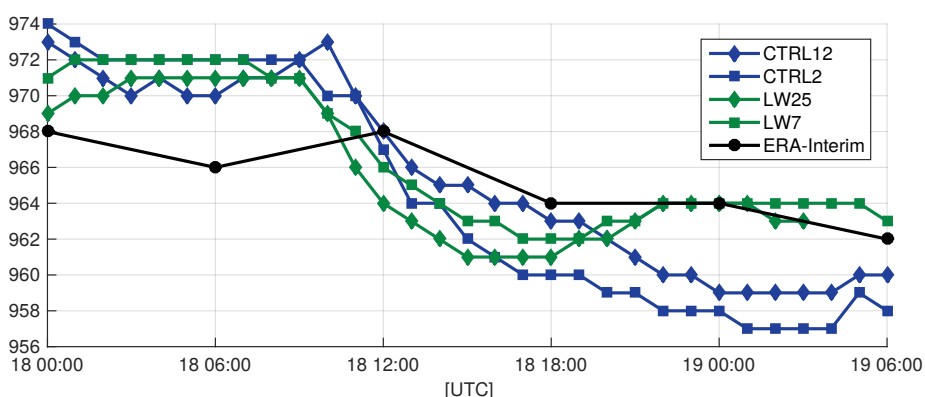

**Figure 5.** Core pressure evolution of the Kyrill II wind storm from 18 January 2007 00 UTC onwards. The black dots represent the 6-hourly ERA-Interim data, the blue diamonds the CTRL12 and the blue squares CTRL2. The green diamonds show the storm core pressure for the 25 km grid-spacing simulation (LW25) performed in Ludwig et al. (2015), and the green squares their simulation with a horizontal grid spacing of 7 km (LW7).







**Figure 6.** Snapshots on 17 January 2007 12 UTC and 18 January 2007 18 UTC. The colored shading indicates the rain-rate [mm/h], the white shading a cloud cover visualization (section 2.4.1), and the white contours geopotential height at 850 hPa [gpdm] using a line spacing of 4 gpdm. (Left) CTRL12 simulation and (right) CTRL2 simulation. The red boxes in the left-hand column denote zoomed areas and the orange box denotes the zoomed area used in figure 7. An animation of this episode is available on the internet (Leutwyler et al., 2015a).





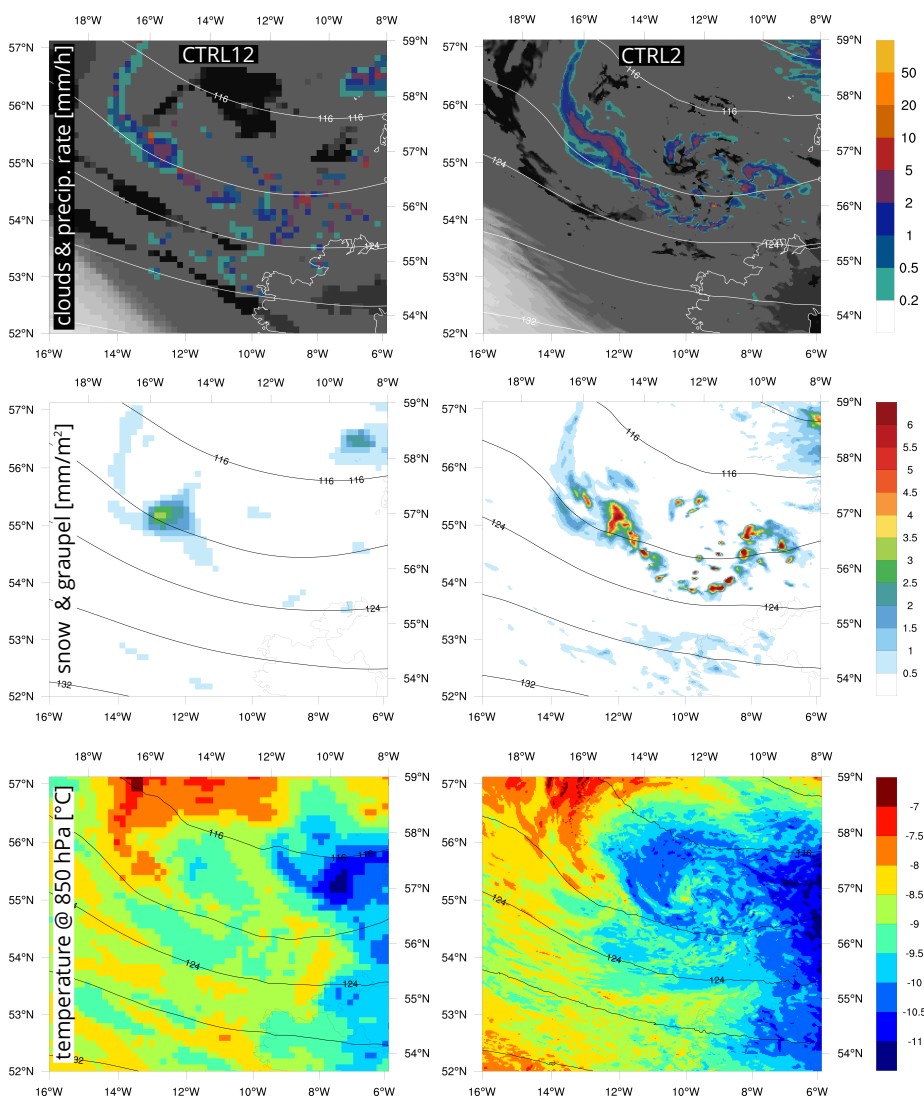

**Figure 7.** Zoomed views in the orange box in figure 6 for (left column) CTRL12 and (right column) CTRL2. (Top) Colored shading indicates the rain-rate [mm/h], the white shading the cloud cover visualization, and the white contours geopotential height at 850 hPa [gpdm], (middle) vertically integrated sum of snow and graupel hydrometeors [mm/m$^2$], (bottom) temperature at 850 hPa [°C].





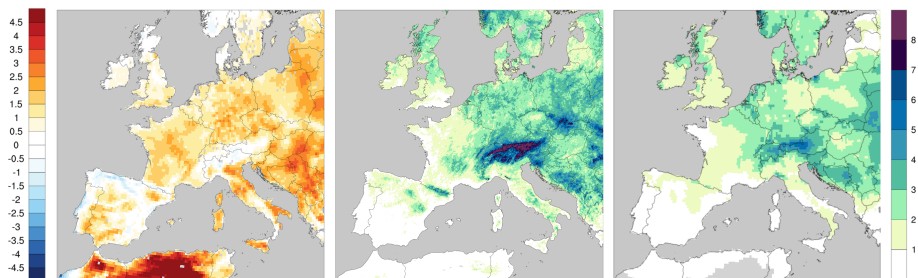

**Figure 8.** Validation of CTRL2 seasonal means: (Left) temperature bias [°K], (middle) simulated precipitation [mm/day], and (right) observations from E-OBS. To account for differences in topography and spatial resolution the model 2m temperature has been height corrected assuming a lapse rate of 0.65 K/100m, before calculating the bias.

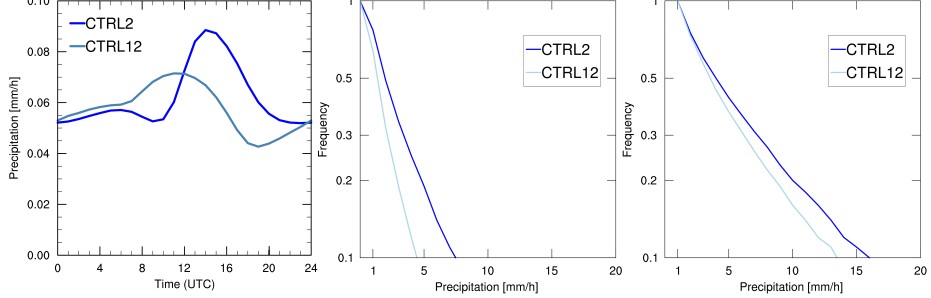

**Figure 9.** (Left) Average diurnal cycle of precipitation over land, (middle) cumulative frequency-intensity distributions of daily-maximum-1h precipitation, and (right) daily precipitation.




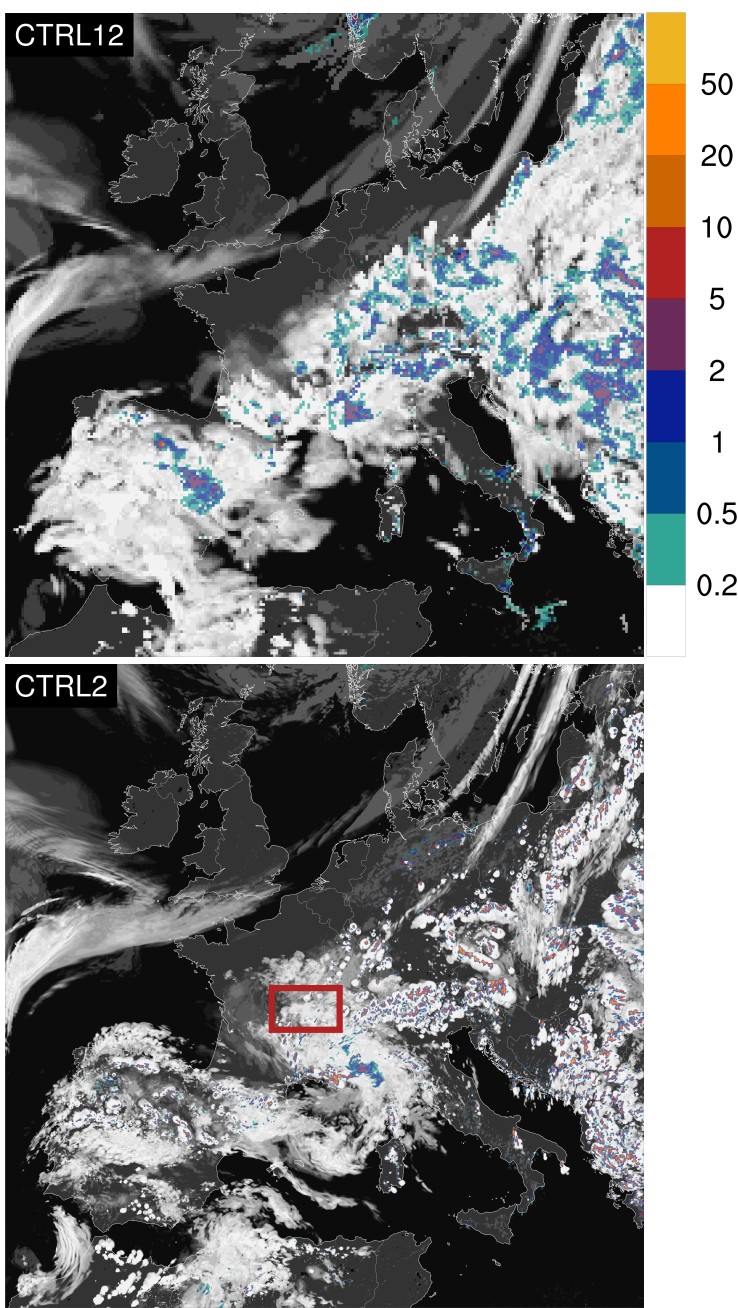

**Figure 10.** Summertime convection over continental Europe. Snapshots on 13 June 2006 12 UTC from (top) CTRL2 and (bottom) CTRL12. The colored shading denotes the 15min-precipitation [mm/h], and the grey shading a cloud visualization. The red box denotes a zoomed area used in figure 7. An animation of this display is available on the internet (Leutwyler et al., 2015b)



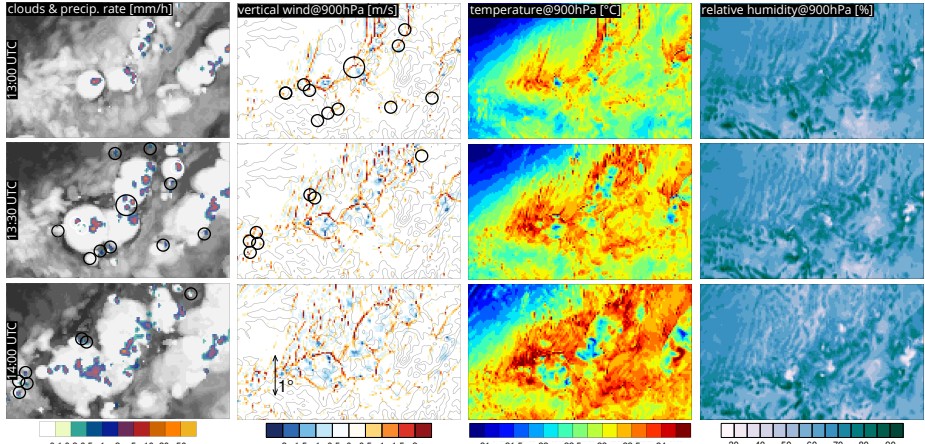

**Figure 11.** Three consecutive snapshots showing (left) precipitation rate [mm/h] and clouds, (left-middle) vertical wind at 900 hPa [m/s] and terrain contours [100 m contour], (right-middle) temperature at 900 hPa [°C], and (right) relative humidity at 900 hPa [%]. The domain corresponds to the red rectangle in figure 10. The black circles in the vertical wind figures denote locations of new convective cells in the next snapshot. In the succeeding snapshot the same convective cells are marked in the left column.

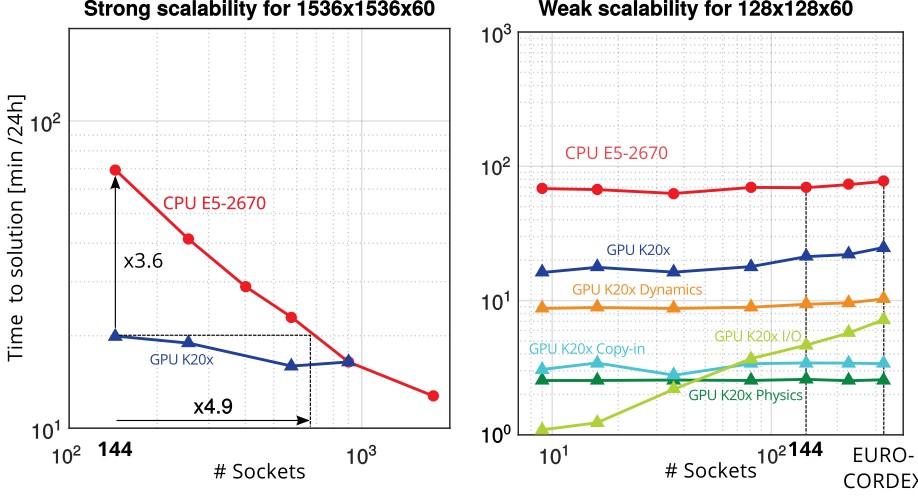

**Figure 12.** Time to solution for a 24h simulation for execution on CPUs architecture (red) and on the same number of GPUs (blue). (Left) Strong scaling for a domain with 1536x1536x60 grid points. The number of sockets is increased while keeping the problem size fixed. (Right) Weak scaling with a per-node size of 128x128x60 grid points. Increasing problem size while keeping the grid points per socket constant. Contributions form several modules: (orange) dynamics, (dark green) physics, (light green) Input/Output, and (light blue) data copy to and from accelerator.