# Peer review of "Towards European-Scale Convection-Resolving Climate Simulations with GPUs: A study with COSMO 4.19"

_Geoscientific Model Development, 2016_

## Short Comment (SC1) · 6 Jun 2016

Dear authors,

In my role as Executive editor of GMD, I would like to bring to your attention our Editorial version 1.1:

http://www.geosci-model-dev.net/8/3487/2015/gmd-8-3487-2015.html

This highlights some requirements of papers published in GMD, which is also available on the GMD website in the 'Manuscript Types' section:

http://www.geoscientific-model-development.net/submission/manuscript_types.html

In particular, please note that for your paper, the following requirements have not been met in the Discussions paper:

[Figure]

- "The main paper must give the model name and version number (or other unique identifier) in the title."

- "If the model development relates to a single model then the model name and the version number must be included in the title of the paper. If the main intention of an article is to make a general (i.e. model independent) statement about the usefulness of a new development, but the usefulness is shown with the help of one specific model, the model name and version number must be stated in the title. The title could have a form such as, "Title outlining amazing generic advance: a case study with Model XXX (version Y)"."

Please name the COSMO model and provide a version number (e.g. COSMO X.Y-gpu) in the title in order to identify the code version this paper is based on. Please correct this upon revision of this article.

Yours,

Astrid Kerkweg

---

## Referee Comment (RC1) · Anonymous Referee #1 · 27 Jun 2016

General Comments:

This is a clearly written manuscript describing world leading efforts to apply high-performance computing to climate studies. With some sharpening of focus, it could make a worthwhile contribution to the literature. Specifically I recommend expanding and deepening section 5, and shortening and making more specific sections 3 and 4.

I suggested minor revisions as my comments and critique should be sufficiently clear for the editor to adjudicate on his/her own, not as an indication of their importance. But also for this reason I would not be inclined to re-review a revised manuscript. This is something the editor can do.

Specific Comments:

[Figure]

1. The conclusion (line 585) makes a concise point of the additional mesoscale structure that becomes representable as the grid is refined to ca 2 km. The manuscript itself is too descriptive and qualitative, and the added value of the 2 km simulations should be more concisely and clearly presented.

2. The most interesting aspect of the manuscript was the proof of concept and computational perspective of the outlook (section 5). Emphasizing the scaling performance, and where and how this might be expected to change either based on different implementations or hardware changes could, when combined with point 1, more sharply guide the community.

Details:

line 8: COSMO (spell out)

line 21: What makes COSMO a climate model? I don't think this is such and appropriate use of the phrase.

line 29: Fine, but I also think this understates what might be an important role for the mesoscale.

line 32: I never think of Dai and Trenberth as a reference for the uncertainties, or approximations required in cumulus parameterization.

line 40: A case could be made for mentioning super-parameterization here

line 50: 'partly resolved' is 'partly' true. What about the stable boundary layer, or clouds that are only a few hundreds of meters, or do you mean this to apply to deep convective systems. More clarity would be helpful.

line 60: "narrow"? Perhaps "fine" would be a better choice.

line 146: How adequate is the Tiedtke references? Is this what COSMO really uses . . . I find it hard to believe that the implementation is unchanged from what Tiedtke describes for implementation at ECMWF.

Line 152: Is it necessary to introduce the DESL acronym?

Line 168: Same goes for GCL

Line 228: I am not sure I would lead by saying how you don't do things.

Line 240: Traditionally abbreviations, such as "srf" in mathematical text are typed in the roman font. Variables take on the italic font. This helps distinguish between "srf" meaning surface, as opposed to 'srf' meaning the entropy times the radius times the free energy... or whatever s*r*f might be.

Line 255: I appreciate your humanity; but describing this as 'sad' might assign unintended value to the situation. For instance, some might argue that 'sad' is too passive a voice. I would simply state that it is estimated but there were X storm related casualties and damage estimates were Y (ref).

Line 272: Check the style guide but I am not used to seeing square brackets to delineate ellipsis.

Line 286: Is "figure 3" a proper name, i.e., "Figure 3"

Line 315: Is there no way to look directly at station data so as to infer the core pressures? I guess this depends on where the core is.

Figure 7: Why not continue to show CTRL 50

Line 346: Is the 'polar low' designation correct? Would be good to be specific and connect better to the literature?

Line 401: But isn't one advantage of the 2km model that it captures orographic influences, and in this case comparing the valley totals to the mean would help quantify this statement?

Line 405: Isn't the ability of convective resolving/permitting simulations to better simulate the diurnal cycle by now an old result?

Line 430: Maybe an earlier Tompkins and Craig reference is warranted in the discussion of cold pools?

Line 460: One should distinguish between weak scaling at fixed resolution and weak scaling more generally. The distinction is that the temporal dimension does not exhibit weak scaling.

Line 490: This discussion would benefit fro some reflection on what the trade offs are of the different ways of comparing things.

Line 500: Why compare 64x64 with 2620, as it forces the reader to do the multiplication. I prefer to call 64x64 4096.

Line 510: Some additional performance metrics, like time per price, or time per power would be informative, even if only qualitative.

Line 530: It may be mention that many CORDEX simulations are performed with very old models (REMO) whose parallel efficiency is quite small.

Line 556: I would prefer to phrase this as: "what does this mean for global simulations"

---

## Referee Comment (RC2) · Anonymous Referee #2 · 5 Jul 2016

The manuscript entitled "Towards European-Scale Convection-Resolving Climate Simulations" describes a new implementation of the COSMO code capable of using GPU cores. In addition, this new implementation is applied by performing two simulations at convection permitting scales over a large domain. These simulations are compared to 12-km ones. Finally, the computational advantages are discussed.

General comments: The manuscript is easy to read and well written. Most figures are clear and the scientific content seems correct. However, the goals of this study are very unclear to me. In the introduction, the authors wrote: "we assess the applicability of the convection-resolving COSMO model on continental scales". I do not often read Geoscientific Model Development but the publication of such an assessment in COSMO technical report seems more relevant to me. If every time somebody is increasing the domain size, he/she publishes a paper, then there would be a lot of useless literature

out there.

About half of this manuscript describes the methodology and the results of two experiments. The result of these simulations are well-established findings: CPSs can model finer structures and more realistic sub-daily statistics of convective precipitation. This was already found in many studies and is not worse being re-communicated, at least not with so many details. A small part of the paper is, in my opinion, relevant for publication, namely Section 5.

I am not sure what recommendation to give for this paper. I think it needs strong revision on the motivation. What do you want to communicate? Where should you publish this communication? I do not think that stating that the COSMO can be used on big domain is a communication relevant for publication in a peer review journal. I know that in the CLM community they also use quite large domains, also at CPS. For example, they are doing big brother experiments with domain of something like 1000 x 1000 grid points. Because, I may not have understood the real motivation of this manuscript, I recommend a major revision. I ask the authors to express the motivation of the paper clearly and to make sure that the communication they want to publish is relevant. In addition, I ask the authors to restructure this paper according to this motivation. Stating that CPSs can model fine structure is most probably not necessary.

Major Comment:

Line 436: the use of a large domain is motivated by the fact that "large domains provides a tool to study cold pools in heterogeneous ...". I am quite sure that the domain size of Ban et al. (2014) or Kendon et al. (2012) are large enough to reproduce these cold pools. In general, in the manuscript, there is no motivation for the use of large domains. Please motivate the need for such large domain.

Minor comments:

L320: You indicate "not shown". Why not using the supplementary material to display

this information?

L415: Typo: logn should be long?

L423: Typo: bahavior should be behavior?

L489: I agree with using socket for the comparison. Still, I think you should provide more information on what is on each sockets. Please describe the types of CPU/GPU that are used. You could also write the energy efficiency of these hardware. This would allow you to provide a rough estimation of the energy saved for a similar simulation in a latter part of the manuscript.

L509: "5 times more sockets". Why using 5 times in the text and 4.9 on the figures. Please be consistent.

---

## Referee Comment (RC3) · A. F. Prein (Referee) · 5 Jul 2016

In the article "Towards European-Scale Convection-Resolving Climate Simulations present an adapted version of the COSMO-CLM model" Leutwyleret al. present a new version of the COSMO-CLM model that allows weather and climate simulations on GPUs. They impressively demonstrate the computational efficiency of this approach and the feasibility of the model to simulate processes from the mesoscale to the synoptic-scale by analyzing a strong, large-scale forced winter storm and weakly forced summertime convective storms. The article is well write well suited for publication in GMD. I have only minor comments that are listed below.

L9: I suggest to replace demonstrate with e.g., present. L9: continental-scale L12: There are several places in the manuscript where commas are missing (here after

"Furthermore,...". Please revise the document accordingly. Here are the locations where I found missing commas but there might be more. L61, 76, 122, 142, 200, 270, 301, 408, 460 L126: You already introduce the acronym COSMO in L109. L149: I would suggest to make Figure 2 to Figure 1 since you mention it first in the text. L274-384: I suggest to shorten this section. For me, the basic message here is that the 12 km simulation performs very similar to the 2 km simulation in this kind of storm but the 2 km simulation is able to add some small-scale processes/details that are not present in the 12 km run. I think this can be communicated more concisely. As you mention at the end of this section, a real comparison and evaluation of the two models is not feasible and therefore the detailed description of the differences between the two models can be shortened. L307: What do you mean with observational reference here? If you refer to ERA-Interim I would change observational to reanalysis. L308: Should be ERA-Interim L330: I cannot see that the band is much narrower in the 2 km model in Fig. 6. Also in the bottom panel I would argue that the 12 and 2 km simulation are remarkably similar and not that the differences are even more pronounced as you state in L303. L340-1: In my printout it is very hard to see the difference between 20 mm/h and 50 mm/h. I have similar issues with Fig. 7, 10, and 11. Changing the color map might help to visualize the differences. L350: "..., does not exhibit much similarity." I would change this to ", are different." L387: I suggest to write "An over-prediction..." L389: Please add which version of E-OBS you used. L397: You could add e.g., the pattern correlation coefficient here to quantify what you mean with well captured. L398: Change observations to observed precipitation L399: You could add Prein and Gobiet 2016 here who compared E-Obs precipitation with high-resolution observations in large areas of your simulation domain. L407: Replace: "is already at its peak around noon" with e.g., "has a precipitation peak at noon". In addition, you state that the convection is still building up in the 2 km model but you do not investigate convection here but only investigate precipitation. L439: ...in the 900 hPa... L556: I would suggest to add that you are talking about AMIP style GCMs here. L576-7: add – between the number and the km as you did in L581:

All Figures: I would suggest adding panel names to your figures (e.g., a,b,c...). It is sometimes hard to know which panel you are referring to in the text. Figure 1: Add Southern to UK Figure 2: right column – Would it be possible to have the contour labels more similar to the other maps. There are only very few labels in these maps. Figure 6: The embedded convection that you are talking about in the text is hard to see in my printouts. A way to visualize this more easily could be to show the vertical velocity (e.g., at 700 hPa) instead of the cloud field. However, I leave it up to you if you want to make this additional effort. Figure 9 right panel: Shouldn't this be mm/d? Figure 10: I guess you mean Figure 12 instead of Figure 7 here? Looking at this snapshots it seems like that the largest differences are found over the Iberian Peninsula and Eastern Europe. The cloud field in these regions look very different in the 2 km simulation. Figure 11: These maps are too small. You might want to split them in two figures or move the maps from the right two columns beneath the left columns.

Best regards, Andreas Prein

Literature: Prein, A. F. and Gobiet, A. (2016), Impacts of uncertainties in European gridded precipitation observations on regional climate analysis. Int. J. Climatol.. doi:10.1002/joc.4706

―――――――――――――

---

## Author Comment (AC2) · 30 Aug 2016

Thank you for your positive and encouraging feedback. The version of COSMO, capable of exploiting GPUs, has been - and still is - a big effort by a large team of contributors and we appreciate that you welcome our work.

*This is a clearly written manuscript describing world leading efforts to apply high-performance computing to climate studies. With some sharpening of focus, it could make a worthwhile contribution to the literature. Specifically I recommend expanding and deepening section 5, and shortening and making more specific sections 3 and 4.*

Similar issues were also raised by Reviewer 3. We revised and shortened the text of sections 3 and 4 and also added more section titles. To shorten the descriptive parts of the manuscript in section 3 and 4, we removed Figure 4 and we moved one of the cases

displayed in former Figure 6 to the supplementary material. Furthermore we removed the panels, displaying relative humidity, in former Figure 11. The corresponding text has also been shortened significantly (most notably Sections 3.1.1 Kyrill Evolution, 3.1.2 Representation of Precipitation along Cold Fronts and 3.1.3 Representation of a Meso-scale Low).

In addition, we substantially expanded section 5:

(1) We clarified the "socket metric" and the implications for the presented experiment (including a new figure, Figure 11 in the revised version). The respective paragraphs now read:

"The full strong-scaling experiment corresponds to a 24 h simulation on a domain of 1536×1536×60 grid points. Input for this simulation consists of the lateral boundary conditions at hourly resolution, amounting to about 120 GB for the whole simulation. Additionally an output workload consisting of about 6 GB is written to the file system. All performance results have been obtained on a heterogeneous Cray XC30 system, located at the Swiss National Supercomputing Centre (CSCS) in Lugano, Switzerland (Piz Daint). The Piz Daint supercomputer consists of a heterogeneous node architecture with an eight-core Intel Xeon E5-2670 CPU and an NVIDIA Tesla K20X GPU per node (Figure 11), and Cray's Aries interconnect using a three-level dragonfly topology to connect the compute nodes. To normalize the performance metrics, they are defined as per socket. In the case of our configuration (Piz Daint, Figure 11), a socket corresponds to either an eight-core Xeon CPU or an NVIDIA K20x GPU.

A socket is the electrical component that provides the connection between the circuit board and the chip sitting on top of it. The advantage of the per-socket metric is its flexibility across architectures, which also allows comparing with individual sockets on a multi-GPU node (fat node). On a fat node, a socket still hosts only a single GPU chip, even if multiple GPU sockets are installed on a PCI express card or on a node. However, for the node configuration found in Piz Daint, this metric is a bit unfair towards

the multi-core systems, since GPUs (today) still need an accompanying CPU hosting the operating system and instructing the GPU. With the socket-based metric, we do not account for that additional CPU. Another metric would be node-to-node comparison, assuming that a node can either consist of one CPU and a GPU, or two CPUs. For such a configuration, the second option would be fairer for the multi-core architecture. In general, node-to-node comparison is useful to compare the various possible node configurations one may find in a supercomputer. However, we believe that for the current study the per-socket performance metric is more useful than node-to-node comparisons, also because nowadays fat-nodes are commercially available."

(2) We elaborate differences in weak-scaling experiments w. r. t. global models by adding the following paragraph: "The weak-scaling approach used here is slightly different for weak-scaling experiments with global simulations, because in these experiments the domain size can not be varied, except by shrinking and expanding the size of the planet . In some global experiments the grid-spacing is varied while keeping the time step constant (Wehner et al., 2011) ."

(3) We discuss potential differences in the weak-scaling behavior w. r. t. the employed numerical schemes in Section 5. 3 (Assessment). We added the following paragraph in the end of the section: "Whether the above assessment can be transferred to GCMs, also depends upon the time-stepping algorithm and its implementation. Many global non-hydrostatic models invoke semi-Lagrangian or spectral approaches (where the total communication costs increases faster than the number of gridpoints). In such cases, perfect weak scaling will be more difficult to achieve than with the current split-explicit scheme."

For further details on these changes, please refer to the revised/marked-up manuscript for the details.

*I suggested minor revisions as my comments and critique should be sufficiently clear for the editor to adjudicate on his/her own, not as an indication of their importance. But*

*also for this reason I would not be inclined to re-review a revised manuscript. This is
something the editor can do.*

**Specific Comments:**

*1. The conclusion (line 585) makes a concise point of the additional mesoscale struc-
ture that becomes representable as the grid is refined to ca 2 km. The manuscript itself
is too descriptive and qualitative, and the added value of the 2 km simulations should
be more concisely and clearly presented.*

Based on your suggestion and the suggestions of Reviewer 3, we added more quanti-
tative information in section 3 (Kyrill case). Besides sharpening the text in general, we
added the following:

(1) We provide an estimate on the width of the cold frontal rain bands:

"In CTRL12, the front is split into successive precipitation bands with maximum pre-
cipitation rates up to 20 mm/h. CTRL2 additionally features small-scale embedded
convection located in the vicinity and along the front, and a more coherent organiza-
tion. The frontal rainbands (precipitation > 5 mm/h) are typically 30-40 km wide in
CTRL12, and substantially narrower in CTRL2 (8-10 km). "

(2) After moving one of the cases in former Figure 6 to the supplementary material,
we added another row of panels for the remaining case (now Fig. 5). The refined
display allows us to establish the representation of narrow cold-frontal rainbands. The
corresponding text reads:

"The distinct narrow cold-frontal rainbands seen in the bottom-right panel of Figure 5
are of distinctly convective origin. They are associated with precipitation rates >20
mm/h, located on the leading edge of the fronts, and aligned with the cold front in
an oblique angle. These systems have been extensively discussed in the literature
(Houze, 2014), and studied using (airborne) radar (e. g. Jorgensen et al. 2003). We
expect differences in location and intensity, due to the ability of CTRL2 to explicitly

resolve the underlying dynamical processes."

Short simulations with atmospheric models in climate mode are of limited predictability and detailed verification of the discussed cases would therefore be dubious. Furthermore, the observational reference of the cases discussed is too weak to allow for detailed verification of the underlying processes.

*2. The most interesting aspect of the manuscript was the proof of concept and computational perspective of the outlook (section 5). Emphasizing the scaling performance, and where and how this might be expected to change either based on different implementations or hardware changes could, when combined with point 1, more sharply guide the community.*

We are very glad that you consider section 5 useful for the community. We substantially expanded it, including a new paragraph on the weak-scaling behavior of global models and added a new figure (current Figure 11). As outlined in more detail above, we clarified the " socket metric" and the implications for the presented experiment. Furthermore we elaborate differences in weak-scaling experiments w. r. t. global models and w. r. t. numerical schemes.

However, the question how the (stencil) implementations scale on other hardware architectures, than those two considered in the current study, is a rather complex topic and would require a dedicated manuscript.

**Details:**

*line 8: COSMO (spell out)*

Changed according to the GMD guidelines.

The sentence now reads: "One of the first atmospheric models that has been fully ported to these architectures is the Consortium for Small-scale Modeling model (COSMO)."

*line 21: What makes COSMO a climate model? I don't think this is such and appropriate use of the phrase.*

We refer to our version of COSMO as a "regional climate model" for two reasons: First, the model includes representations of the couplings with the land-surface, in particular with soil moisture and snow cover. Second, the model enables simulations over climatological time periods (at least a decade), and can thus be validated not only in an NWP context, but also with respect to climatological distributions. On a more technical level, for climate-scale simulations with COSMO, several enhancements not needed in NWP mode, such as a time-dependent leaf-area index or time-dependent sea surface temperatures, are needed. Many of these additional climate features are maintained by the COSMO-CLM community. However, since the COSMO v4.19 is not an official CLM version, we refrain from calling our model version COSMO-CLM for now.

The sentence has been changed to: "With the COSMO model, we now use a weather and climate model that has all the necessary modules required for real-case convection-resolving regional climate simulations on GPUs."

*line 29: Fine, but I also think this understates what might be an important role for the mesoscale.*

We also think that convection-resolving models provide substantial insight about meso-scale processes. It should therefore not miss out here.

We modified the first paragraph accordingly. It now reads: "The inadequate representation of clouds and moist convection represents a major challenge of state-of-the-art climate models (Stevens and Bony, 2013). An important component of the problem are the scale interactions between small-scale turbulent and convective processes at scales around and below 1 km, and larger-scale/meso-scale weather systems at scales around O(10 km-1000 km). Within these scale interactions, individual convective cells may organize into meso-scale weather systems such as squall lines or meso-scale

convective systems. Current global and regional climate models typically operate at grid spacings on the order of 10-300 km, and are thus unable to explicitly represent many of these interactions."

*line 32: I never think of Dai and Trenberth as a reference for the uncertainties, or approximations required in cumulus parameterization. line 40: A case could be made for mentioning super-parameterization here*

We would like to address these two issues together by citing Randall et al. (2003) instead. They provide a nice overview of cloud-related uncertainties, and propose the super-parametrization concept.

*line 50: 'partly resolved' is 'partly' true. What about the stable boundary layer, or clouds that are only a few hundreds of meters, or do you mean this to apply to deep convective systems. More clarity would be helpful.*

Yes, some clarifications could help indeed. We changed partly resolved to underresolved and added further explanations.

The first part of the paragraph now reads: "While the convection-resolving approach shows very promising results, turbulent and convective motions are still underresolved (Wyngaard, 2004). Grid spacings of O(1 km) are comparable to the size of the particularly energetic convective eddies in the planetary boundary layer (Zhou et al., 2014). At this resolution, shallow clouds still need to be parametrized and deep-convective clouds tend to be too large, too laminar, too vicious and too widely spaced apart (Clark et al., 2016). Using numerical simulations of an idealized squall line, Bryan et al. (2003) showed ..."

*line 60: "narrow"? Perhaps "fine" would be a better choice.*

Yes, it is. Changed accordingly.

*line 146: How adequate is the Tiedtke references? Is this what COSMO really uses . . . I find it hard to believe that the implementation is unchanged from what Tiedtke*

*describes for implementation at ECMWF.*

We think it is still adequate to quote the Tiedtke (1989) reference, although the version used in COSMO has indeed been adapted from the original implementation. Some of these changes are described in the COSMO documentation, some are fairly recent and have not yet been documented thoroughly.

*Line 152: Is it necessary to introduce the DESL acronym? Line 168: Same goes for GCL*

We removed these acronyms form the manuscript, since they are indeed not used later on.

*Line 228: I am not sure I would lead by saying how you don't do things.*

Based on the suggestion, we substantially altered the structure of the respective sub-section. Please see the revised version of the manuscript.

*Line 240: Traditionally abbreviations, such as "srf" in mathematical text are typed in the roman font. Variables take on the italic font. This helps distinguish between "srf"meaning surface, as opposed to 'srf' meaning the entropy times the radius times the free energy. . . or whatever s\*r\*f might be.*

We agree and have consulted the SI brochure 8 (mentioned on the GMD website). We changed the symbols accordingly.

*Line 255: I appreciate your humanity; but describing this as 'sad' might assign unintended value to the situation. For instance, some might argue that 'sad' is too passive a voice. I would simply state that it is estimated but there were X storm related casualties and damage estimates were Y (ref).*

We decided to remove this sentence from the manuscript. We chose the Kyrill case because it has been well assessed by Fink et al. (2009) and Ludwig et al. (2015). Actually, we do not assess the impact of the storm itself.

*Line 272: Check the style guide but I am not used to seeing square brackets to delineate ellipsis.*

We found no recommendation in the GMD guides, but it seems common practice to use round brackets. We changed it here and in the rest of the manuscript accordingly.

*Line 286: Is "figure 3" a proper name, i.e., "Figure 3"*

Yes. We changed all the occurrences of that term accordingly.

*Line 315: Is there no way to look directly at station data so as to infer the core pressures? I guess this depends on where the core is.*

While composing this manuscript we tried to find station data. For instance, we also tried to infer station data from the Synop- and Bodenwetterkarten of DWD. At least in our sources, there are not sufficient stations available that are located close enough to the storm core, even after it made landfall.

*Figure 7: Why not continue to show CTRL 50*

Good idea. We have added panels with results from CTRL50 and adapted the text accordingly (this is now in Fig.6, as we have dropped the previous Fig.4). At the same time we also reduced visual clutter in the figure, stemming from the coordinate labels. At this occasion we also corrected a small bug in the display: In the initial version, smoothing was applied to the geopotential height contours in the panels of precipitation and temperature, which was removed. This bugfix makes the geopt. contours slightly noisy, but makes all three rows of the display consistent, and does not affect the conclusions drawn from the analysis.

*Line 346: Is the 'polar low' designation correct? Would be good to be specific and connect better to the literature?*

When composing the initial version of this manuscript we used the following definition: "A polar low is a small, but fairly intense maritime cyclone that forms poleward of the

main baroclinic zone (the polar front or other major baroclinic zone). The horizontal scale of the polar low is approximately between 200 and 1000 kilometres and surface winds near or above gale force."

Erik A. Rasmussen and John Turner (2003). *Polar Lows*. doi: 10.1017/CBO9780511524974

Actually we do not assess the relevant forcing mechanisms in detail and for this study it is not particularly relevant if it this particular eddy can be classified as polar low or not. We therefore replaced the term "polar low" by "meso-scale vortex", which is a less restrictive decision.

*Line 401: But isn't one advantage of the 2km model that it captures orographic influences, and in this case comparing the valley totals to the mean would help quantify this statement?*

It has indeed been shown that the added detail, provided by the increased resolution, improves the representation of orographic influence (see Ban et al. 2015, and Prein et al., 2015 for a review). As explained in the text and the references therein, the low station density and uneven distribution introduces substantial uncertainties in areas with high spatial variability. The sampling uncertainty will manifest itself as long as (low resolution) gridded data is used. Direct validation with station data could help circumvent this problem

The goal of Figure 8 (revised version) is to validate the continental scale pattern. Thoroughly validating valley and mountain-peak totals would involve using station data and high-resolution data sets. A study with a more local-scale focus would be better suited for such a validation. Moreover, as you suggested in the introductory comment, the particular section in the manuscript should be shortened rather than extended.

*Line 405: Isn't the ability of convective resolving/permitting simulations to better simulate the diurnal cycle by now an old result?*

Yes. That is why we state the following a few lines below:

L410: "It has previously been shown for smaller domains (Kendon et al., 2012; Ban et al., 2014) that the behavior of the convection-resolving model fits observation much better. Our results are qualitatively consistent with these studies, although the differences in daily precipitation statistics are larger for our simulation."

A shortcoming of the simulation used in the address section of the manuscript is that it is only a few months long. It would be therefore be questionable to conduct detailed validation on local scales using observational precipitation data sets. Therefore, we mostly perform model-to-model comparisons. Our concept is to use well established results, from simulations on smaller domains (e. g. Hohenegger et al. 2009 or Ban et al., 2014), to show that their findings also apply to simulations on European-scale domains.

*Line 430: Maybe an earlier Tompkins and Craig reference is warranted in the discussion of cold pools?*

It definitely is. We think that Tompkins (2001) should be suited best.

*Line 460: One should distinguish between weak scaling at fixed resolution and weak scaling more generally. The distinction is that the temporal dimension does not exhibit weak scaling.*

That could indeed be useful information for the reader, especially since weak-scaling experiments in regional and global models are typically performed slightly differently. With a global model, an experiment similar to the one performed here, would be shrinking and expanding the size of the planet while proportionally adapting the number of nodes used. However, weak-scaling experiments are rarely performed this way. Instead, typically the resolution is decreased, while keeping the time step constant. When performing a weak-scaling experiment with a regional model, the grid spacing, time step and the time to solution can remain the same across all domain sizes. For

strong scaling, on the other hand, only the number of grid points per node changes, while the time dimension cannot be scaled.

To clarify the different approaches, we adapted the first part of the paragraph the following way:

"What are the computational requirements to perform a convection-resolving simulation on the European scale? Here we restrict the analysis to two key performance metrics:

1. Strong scaling: The achievable time to solution for a fixed simulation domain, fixed grid spacing and domain size, while increasing the computational resources. For linear scaling, the time to solution will increase inverse proportionally to the computational resources, allocated to the problem. Here the time step (which is constrained by the grid spacing through the Courant stability criterion) can be kept constant and hence the computational task has a fixed size

2. Weak scaling: The achievable time to solution when the domain size is increased proportionally with the computational resources, while keeping the grid spacing and the time step fixed. For linear scaling, the time to solution would remain the same for all domain sizes.

The weak-scaling approach used here is slightly different for weak-scaling experiments with global simulations, because in these experiments the domain size can not be varied, except by shrinking and expanding the size of the planet. In some global experiments the grid-spacing is varied while keeping the time step constant at the value required by the simulation with the finest grid (e. g. Wehner at al., 2011)."

*Line 490: This discussion would benefit from some reflection on what the trade offs are of the different ways of comparing things.*

This point was also raised by other reviewers. We added further explanations and it now reads:

"(...) To normalize the performance metrics, they are defined as per socket. In the case of our configuration (Piz Daint, Figure 11), a socket corresponds to either an eight-core Xeon CPU or an NVIDIA K20x GPU.

A socket is the electrical component that provides the connection between the circuit board and the chip sitting on top of it. The advantage of the per-socket metric is its flexibility across architectures, which also allows comparing with individual sockets on a multi-GPU node (fat node). On a fat node, a socket still hosts only a single GPU chip, even if multiple GPU sockets are installed on a PCI express card or on a node. However, for the node configuration found in Piz Daint, this metric is a bit unfair towards the multi-core systems, since GPUs (today) still need an accompanying CPU hosting the operating system and instructing the GPU. With the socket-based metric, we do not account for that additional CPU. Another metric would be node-to-node comparison, assuming that a node can either consist of one CPU and a GPU, or two CPUs. For such a configuration, the second option would be fairer for the multi-core architecture. In general, node-to-node comparison is useful to compare the various possible node configurations one may find in a supercomputer. However, we believe that for the current study the per-socket performance metric is more useful than node-to-node comparisons, also because nowadays fat-nodes are commercially available."

*Line 500: Why compare 64x64 with 2620, as it forces the reader to do the multiplication. I prefer to call 64x64 4096.*

Indeed. We changed that number according to your suggestion.

*Line 510: Some additional performance metrics, like time per price, or time per power would be informative, even if only qualitative.*

The issues raised here are among the key questions when pushing towards global kilometer-scale models. While a qualitative result would indeed be very useful to scientists in the atmospheric domain, others would probably not appreciate us estimating these numbers in an ad-hoc way. Energy to solution experiments have to be carried

out with great care and would require designing a dedicated experiment and study.

A video from an online presentation at SC13 from Thomas Schulthess et al., which includes an analysis of energy to solution can be found here: https://www.youtube.com/watch?v=X5PqyfXc9pAt=21m20s

*Line 530: It may be mention that many CORDEX simulations are performed with very old models (REMO) whose parallel efficiency is quite small.*

We hesitate to criticize other models and modeling groups without performing a thorough performance assessment ourselves.

*Line 556: I would prefer to phrase this as: "what does this mean for global simulations"*

We changed the term accordingly.
* * *

---

## Author Comment (AC3) · 30 Aug 2016

Thank you for your review. Since the report mostly consists of a general critique, we chose to individually address the points raised, where appropriate.

*The manuscript entitled "Towards European-Scale Convection-Resolving Climate Simulations" describes a new implementation of the COSMO code capable of using GPU cores. In addition, this new implementation is applied by performing two simulations at convection permitting scales over a large domain. These simulations are compared to 12-km ones. Finally, the computational advantages are discussed.*

**General comments:** *The manuscript is easy to read and well written. Most figures are clear and the scientific content seems correct.*

[Figure]

We are glad that the scientific correctness of the presented results is acknowledged.

*However, the goals of this study are very unclear to me. In the introduction, the authors wrote: "we assess the applicability of the convection-resolving COSMO model on continental scales". I do not often read Geoscientific Model Development but the publication of such an assessment in COSMO technical report seems more relevant to me. If every time somebody is increasing the domain size, he/she publishes a paper, then there would be a lot of useless literature out there.*

In our view the paper fits very well into the scope of the journal. The website states that GMD is dedicated to the publication of "the description, development, and evaluation of numerical models of the Earth system and its components". Further description is available regarding the specific manuscript type applicable to our paper (model evaluation paper, see http://www.geoscientific-model-development.net/about/manuscript_types.html): "Model evaluation is an important component of most GMD papers. Model development papers in particular often include a large proportion of evaluation".

The description also clarifies the level of evaluation: "It is, however, common for pure evaluation papers to contain substantial conclusions about geoscience rather than about models, and such papers are not suitable for submission to GMD. [...]."

According to these descriptions, our manuscript fits the appropriate manuscript type very well.

The Reviewer also states: *"If every time somebody is increasing the domain size, he/she publishes a paper, then there would be a lot of useless literature out there".*

We consider this statement a bit unfriendly and would like to reply as follows: (1) The current paper is beyond increasing the domain size. Actually, it is the first paper to demonstrate the feasibility of a fully GPU-based model to conduct regional climate experiments. (2) The review makes it sound as if we would increase the domain merely

by a little bit. However, we are actually expanding the previous domain of Ban et al. (2015) by about an order of magnitude (in area). The current version enables studying the climate of a continent, the previous that of an intermediate mountain range.

*About half of this manuscript describes the methodology and the results of two experiments. The result of these simulations are well-established findings: CPSs can model finer structures and more realistic sub-daily statistics of convective precipitation. This was already found in many studies and is not worse being re-communicated, at least not with so many details. A small part of the paper is, in my opinion, relevant for publication, namely Section 5. [...]*

It is correct that some of the results qualitatively agree with recent existing literature (Kendon et al. 2012, Ban et al. 2014, Prein et al. 2015). However, we are using an entirely different code version (with a completely rewritten dynamical core), and a computational domain that is at least 10 times as large as previous high-resolution studies over Europe. We think it is useful to corroborate the previous results and demonstrate the suitability of the GPU-based large-domain approach. Furthermore, our paper also demonstrates the ability of non-hydrostatic models to represent (1) wrap-up of small vorticies, (2) narrow convective cold-frontal rainbands (section 3) as well as (3) propagating cold-air pools and gust fronts associated with thunderstorm outflows (section 4). In fact, these differences not only represent an increase in model detail due to the increased resolution, but also changes in physical behavior. We are not aware of any detailed descriptions of such features in high-resolution simulations spanning the European continent.

That said, it is correct that Section 5 is particularly essential to the paper. Following suggestions of Referees 1 and 3, we have expanded section 5: (1) We clarified the " socket metric" and the implications for the presented experiment (including a new figure, Fig. 11 in the revised version). (2) We elaborate differences in weak-scaling experiments w. r. t. global simulations and (3) w. r. t. to the employed numerical schemes.

Furthermore we made some shortening in sections 3. We have also deleted a figure (former Fig. 4), and reduced the number of panels in another one (current Fig. 10). We think that with these shortenings the level of detail in the presentation is fine for a GMD article.

*I am not sure what recommendation to give for this paper. I think it needs strong revision on the motivation. What do you want to communicate? Where should you publish this communication? I do not think that stating that the COSMO can be used on big domain is a communication relevant for publication in a peer review journal. [...]*

Again, the review makes it sound as if we would merely increase the domain by just a small little bit. The details are as follows: The computational domain in our experiments (1536x1536x60 grid points) is an order of magnitude larger than what has been established and used in long-term RCM simulations (500x500x60 grid points by Ban et al., 2014). Increasing the problem size by an order of magnitude requires thorough and continued (re-)evaluation of the model capabilities. We think that using and reproducing established results in the process is a useful approach.

Besides confirming the applicability of the model on continental-scale domains, we present a number of rather novel results and ideas. For instance we are currently unaware of (peer-reviewed) publications of:

1. A weather and climate model which is able to execute the entire time stepping algorithm on GPU accelerators.

2. An assessment of the computational performance of such a model with full model physics.

3. A week-long convection-resolving real-case simulation of an entire extratropical synoptic systems (winter storm Kyrill).

4. The cloud visualization technique presented in section 2.4.1.
5. The demonstrationof our model's ability to scale weakly over very large domains.

*I know that in the CLM community they also use quite large domains, also at CPS. For example, they are doing big brother experiments with domain of something like 1000 x 1000 grid points.*

The Reviewer's comment is very unspecific. We are unaware of a publication about these efforts.

*Because, I may not have understood the real motivation of this manuscript, I recommend a major revision. I ask the authors to express the motivation of the paper clearly and to make sure that the communication they want to publish is relevant. In addition, I ask the authors to restructure this paper according to this motivation. Stating that CPSs can model fine structure is most probably not necessary.*

In revising the paper, we have partly followed these recommendations.

*Line 436: the use of a large domain is motivated by the fact that "large domains provides a tool to study cold pools in heterogeneous ...". I am quite sure that the domain size of Ban et al. (2014) or Kendon et al. (2012) are large enough to reproduce these cold pools. In general, in the manuscript, there is no motivation for the use of large domains. Please motivate the need for such large domain.*

We removed "large domains" from the particular sentence. It now reads: "The use of high-resolution models provides a tool to study cold pools in heterogeneous areas. Here we focus on the subdomain indicated . . ."

Furthermore added a concise motivation for using large domains in Section 2:

"The analysis domain excludes grid columns close to or within the relaxation zone (50 km distance to the CTRL2 boundary) and contains 1536x1536 grid points (2900x2900 km$^2$). It should therefore be large enough for small-scale processes to fully develop (Leduc and Laprise, 2009; Brisson et al., 2016)."

And we extended the paragraph in the end of Section 6 with the following sentences:

"Once established, such simulation capabilities will enable investigations of continental-scale climate feedbacks, sensitive to the treatment of deep convection, or assembling model-climatologies of interactions between convective meso/small-scale and synoptic-scale systems."

**Minor comments:**

*L320: You indicate "not shown". Why not using the supplementary material to display this information?*

We added the domain-mean of the precipitation rate at 18 UTC: CTRL12: 2.7 mm/day, CTRL2: 3.55 mm/day. This is the same snapshot as in Figure 6 (top row).

*L415: Typo: logn should be long? L423: Typo: bahavior should be behavior?*

We fixed these spelling mistakes.

*L489: I agree with using socket for the comparison. Still, I think you should provide more information on what is on each sockets. Please describe the types of CPU/GPU that are used. You could also write the energy efficiency of these hardware. This would allow you to provide a rough estimation of the energy saved for a similar simulation in a latter part of the manuscript.*

The paragraph has been criticized by the other reviewers as well. Therefore we added further clarifications, including a new figure (currently Figure 11). The sections now read:

"The full strong-scaling experiment corresponds to a 24 h simulation on a domain of 1536×1536×60 grid points. Input for this simulation consists of the lateral boundary conditions at hourly resolution, amounting to about 120 GB for the whole simulation. Additionally an output workload consisting of about 6 GB is written to the file system. All performance results have been obtained on a heterogeneous Cray XC30 system,

located at the Swiss National Supercomputing Centre (CSCS) in Lugano, Switzerland (Piz Daint). The Piz Daint supercomputer consists of a heterogeneous node architecture with an eight-core Intel Xeon E5-2670 CPU and an NVIDIA Tesla K20X GPU per node (Figure 11), and Cray's Aries interconnect using a three-level dragonfly topology to connect the compute nodes. To normalize the performance metrics, they are defined as per socket. In the case of our configuration (Piz Daint, Figure 11), a socket corresponds to either an eight-core Xeon CPU or an NVIDIA K20x GPU.

A socket is the electrical component that provides the connection between the circuit board and the chip sitting on top of it. The advantage of the per-socket metric is its flexibility across architectures, which also allows comparing with individual sockets on a multi-GPU node (fat node). On a fat node, a socket still hosts only a single GPU chip, even if multiple GPU sockets are installed on a PCI express card or on a node. However, for the node configuration found in Piz Daint, this metric is a bit unfair towards the multi-core systems, since GPUs (today) still need an accompanying CPU hosting the operating system and instructing the GPU. With the socket-based metric, we do not account for that additional CPU. Another metric would be node-to-node comparison, assuming that a node can either consist of one CPU and a GPU, or two CPUs. For such a configuration, the second option would be fairer for the multi-core architecture. In general, node-to-node comparison is useful to compare the various possible node configurations one may find in a supercomputer. However, we believe that for the current study the per-socket performance metric is more useful than node-to-node comparisons, also because nowadays fat-nodes are commercially available."

Energy to solution experiments would require its own dedicated study. An experiment, using a similar version of COSMO, was performed by T. Schulthess et al. on the Piz Daint supercomputer. See here for an online presentation at SC13, including an analysis of energy to solution: https://www.youtube.com/watch?v=X5PqyfXc9pAt=21m20s

*L509: "5 times more sockets". Why using 5 times in the text and 4.9 on the figures. Please be consistent.*

[Figure]

Rounding the number 4.9 to 5 is indeed inconsistent with Figure 12. We changed that number in the text.

---

## Author Comment (AC4) · 30 Aug 2016

Thank you for your detailed review of our manuscript and the very useful feedback. We appreciate the time you have invested in this review.

*In the article "Towards European-Scale Convection-Resolving Climate Simulations present an adapted version of the COSMO-CLM model" Leutwyler et al. present a new version of the COSMO-CLM model that allows weather and climate simulations on GPUs. They impressively demonstrate the computational efficiency of this approach and the feasibility of the model to simulate processes from the mesoscale to the synoptic-scale by analyzing a strong, large-scale forced winter storm and weakly forced summertime convective storms. The article is well write well suited for publication in GMD. I have only minor comments that are listed below.*

**Minor Comments:**

*L9: I suggest to replace demonstrate with e.g., present.*

We changed the term according to your suggestion.

*L9: continental-scales*

This term consists of an adjective and a noun and does not constitute a compound adjective.

*L12:There are several places in the manuscript where commas are missing (here after Furthermore,...". Please revise the document accordingly. Here are the locations where I found missing commas but there might be more. L61, 76, 122, 142, 200, 270, 301, 408, 460*

Thanks for checking all those commas. We added an additional comma in L301. However, we prefer not to put commas after conjunctive adverbs, except if they imply contradiction.

*L126: You already introduce the acronym COSMO in L109.*

Good catch, thanks.

*L149: I would suggest to make Figure 2 to Figure 1 since you mention it first in the text.*

We switched them according to your suggestion.

*L274-384: I suggest to shorten this section. For me, the basic message here is that the 12 km simulation performs very similar to the 2 km simulation in this kind of storm but the 2 km simulation is able to add some small-scale processes/details that are not present in the 12 km run. I think this can be communicated more concisely. As you mention at the end of this section, a real comparison and evaluation of the two models is not feasible and therefore the detailed description of the differences between the two models can be shortened.*

Yes, the basic message is that the two models agree on the large and meso-alpha-scale. The important differences between the models for the storm case are pointed out later in the manuscript. As suggested, we substantially shortened this section (see revised manuscript) and removed a figure (former Figure 4).

*L307: What do you mean with observational reference here? If you refer to ERA-Interim I would change observational to reanalysis.*

The point is that we have tried to find station observations that would allow deriving the core pressure development of Kyrill II. However, we couldn't find enough stations that were located close enough to the storm core.

We have adapted the sentence. It now reads:

"It should be noted that the observational reference from measurement stations and balloon soundings is rather weak, as this was a rapidly developing small-scale cyclone."

*L308: Should be ERA-Interim*

Changed according to your suggestion.

*L330: I cannot see that the band is much narrower in the 2 km model in Fig. 6. Also in the bottom panel I would argue that the 12 and 2 km simulation are remarkably similar and not that the differences are even more pronounced as you state in L303.*

In our effort to shorten sections 3 and 4, we moved the first case displayed in Fig. 6 to the supplementary material. This action allowed providing an additional row of panels for the Kyrill II case. In the new panels, the differences in rainband width are more evident. Furthermore we estimated their width (counting gridpoints by hand).

We have added the following sentences in Section 3.1.2:

"In CTRL12, the front is split into successive precipitation bands with maximum precipitation rates up to 20 mm/h. CTRL2 additionally features small-scale embedded convection located in the vicinity and along the front, and a more coherent organiza-

tion. The frontal rainbands (precipitation > 5 mm/h) are typically 30-40 km wide in CTRL12, and substantially narrower in CTRL2 (8-10 km).)"

Furthermore we expanded the discussion on the narrow frontal rainbands just below:

"The distinct narrow cold-frontal rainbands seen in the bottom-right panel of Figure 5 are of distinctly convective origin. They are associated with precipitation rates >20 mm/h, located on the leading edge of the fronts, and aligned with the cold front in an oblique angle. These systems have been extensively discussed in the literature (Houze, 2014), and studied using (airborne) radar (e. g. Jorgensen et al. 2003). We expect differences in location and intensity, due to the ability of CTRL2 to explicitly resolve the underlying dynamical processes"

The differences in physical behavior should now be more evident.

*L340-1: In my printout it is very hard to see the difference between 20 mm/h and 50 mm/h. I have similar issues with Fig. 7, 10, and 11. Changing the color map might help to visualize the differences.*

The main problem is that the grid spacing of CTRL2 is finer than the resolution of current computer screens (except for the really expensive ones) and standard printers. To increase the visibility of the details and small-scale structures, we have added another row of zoomed panels in Figure 5.

*L350: "..., does not exhibit much similarity." I would change this to ", are different."*

Changed according to your suggestions. The sentence now reads:

"However, while the geopotential height contours compare rather well, the associated precipitation pattern is different."

*L387: I suggest to write "An over-prediction..."*

Changed according to your suggestions. The sentence now reads: "An over prediction of summer temperature is a long standing issue for COSMO-CLM and other RCMs, ..."

*L389: Please add which version of E-OBS you used.*

We have added the version according to your suggestion.

*L397: You could add e.g., the pattern correlation coefficient here to quantify what you mean with well captured.*

We would like to omit detailed statistical analysis (for now), since the simulated period is still rather short (3 months).

*L398: Change observations to observed precipitation*

Changed according to your suggestion.

*L399: You could add Prein and Gobiet 2016 here who compared E-Obs precipitation with high-resolution observations in large areas of your simulation domain.*

We were not aware that this paper got published by now. Added according to the suggestion.

*L407: Replace: "is already at its peak around noon" with e.g., "has a precipitation peak at noon". In addition, you state that the convection is still building up in the 2 km model but you do not investigate convection here but only investigate precipitation.*

We like the "is already at its peak around noon" formulation and would like to retain it. However, we changed the term "convection" to "precipitation".

*L439: ...in the 900 hPa...*

Thanks. Changed according to your suggestions.

*L556: I would suggest to add that you are talking about AMIP style GCMs here.*

Based on the suggestion of Reviewer 1, we have changed this sentence to: "What does this mean for global simulations."

*L576-7: add – between the number and the km as you did in L581:*

Changed according to your suggestions.

*All Figures: I would suggest adding panel names to your figures (e.g., a,b,c...). It is sometimes hard to know which panel you are referring to in the text.*

We've identified Figures 8 and 9 to be particularly confusing. We have added labels to these panels, according to your suggestion.

*Figure 1: Add Southern to UK*

We have added "Southern" according to your suggestion.

*Figure 2: right column – Would it be possible to have the contour labels more similar to the other maps. There are only very few labels in these maps.*

We guess you meant Figure 3 (Figure 2 in the initial manuscript doesn't have columns) in the original manuscript. Yes more labels could be helpful. We added more contour-labels in the revised version of Figure 3, according to your suggestion.

*Figure 6: The embedded convection that you are talking about in the text is hard to see in my printouts. A way to visualize this more easily could be to show the vertical velocity (e.g., at 700 hPa) instead of the cloud field. However, I leave it up to you if you want to make this additional effort.*

We moved the first case displayed in this panel to the supplementary material and added an additional zoom level for the remaining case. This allowed us to describe the remaining case with more detail. As an example for the convective motions associated with cold-fronts, we introduced reference to narrow cold frontal rain bands.

*Figure 9 right panel: Shouldn't this be mm/d?*

Yes it should. Changed accordingly.

*Figure 10: I guess you mean Figure 12 instead of Figure 7 here? Looking at this snapshots it seems like that the largest differences are found over the Iberian Peninsula*

*and Eastern Europe. The cloud field in these regions look very different in the 2 km simulation.*

We indeed caught the wrong labels. Good catch. Thanks.

*Figure 11: These maps are too small. You might want to split them in two figures or move the maps from the right two columns beneath the left columns.*

In our effort to shorten sections 3 and 4, we removed the panels displaying relative humidity. We hope that the remaining panels are now large enough.

*Best regards, Andreas Prein*
*Literature: Prein, A. F. and Gobiet, A. (2016), Impacts of uncertainties in European gridded precipitation observations on regional climate analysis. Int. J. Climatol.. doi:10.1002/joc.4706*

<hr style="width:40%;text-align:left;margin-left:0">